# Identification of the kinase STK25 as an upstream activator of LATS signaling

Sanghee Lim[1], Nicole Hermance[2], Tenny Mudianto[1,3], Hatim M. Mustaly[1], Ian Paolo Morelos Mauricio[1], Marc A. Vittoria[1], Ryan J. Quinton[1], Brian W. Howell [4], Hauke Cornils[5], Amity L. Manning[2] & Neil J. Ganem[1,6]

The Hippo pathway maintains tissue homeostasis by negatively regulating the oncogenic transcriptional co-activators YAP and TAZ. Though functional inactivation of the Hippo pathway is common in tumors, mutations in core pathway components are rare. Thus, understanding how tumor cells inactivate Hippo signaling remains a key unresolved question. Here, we identify the kinase STK25 as an activator of Hippo signaling. We demonstrate that loss of STK25 promotes YAP/TAZ activation and enhanced cellular proliferation, even under normally growth-suppressive conditions both in vitro and in vivo. Notably, STK25 activates LATS by promoting LATS activation loop phosphorylation independent of a preceding phosphorylation event at the hydrophobic motif, which represents a form of Hippo activation distinct from other kinase activators of LATS. *STK25* is significantly focally deleted across a wide spectrum of human cancers, suggesting *STK25* loss may represent a common mechanism by which tumor cells functionally impair the Hippo tumor suppressor pathway.

[1] The Laboratory of Cancer Cell Biology, Department of Pharmacology and Experimental Therapeutics, Boston University School of Medicine, Boston, MA 02118, USA. [2] Department of Biology and Biotechnology, Worcester Polytechnic Institute, Worcester, MA 01605, USA. [3] Department of Medical Oncology, Dana-Farber Cancer Institute, Harvard Medical School, Boston, MA 02215, USA. [4] Department of Neuroscience and Physiology, SUNY Upstate Medical University, Syracuse, NY 13210, USA. [5] Evotec, Hamburg 22419, Germany. [6] Division of Hematology and Oncology, Department of Medicine, Boston University School of Medicine, Boston, MA 02118, USA. Correspondence and requests for materials should be addressed to N.J.G. (email: nganem@bu.edu)

First discovered in *Drosophila* as a regulator of organ size, the Hippo tumor suppressor pathway has emerged as a key actor in maintaining tissue homeostasis through the regulation of cell proliferation and survival[1]. The key mediators of Hippo signaling are LATS1 and LATS2 (large tumor suppressor) kinases, which function to negatively regulate the activity of the oncogenic transcriptional co-activators Yes-associated protein (YAP) and transcriptional co-activator with PDZ-binding motif (TAZ)[2,3]. Upon stimulation of Hippo signaling, activated LATS kinases directly phosphorylate YAP/TAZ at conserved serine residues, promoting YAP/TAZ nuclear extrusion and subsequent degradation[3]. By contrast, in the absence of LATS activation, YAP/TAZ translocate into the nucleus, where they bind to the TEAD/TEF family of transcription factors to promote expression of genes essential for proliferation and survival[4–6]. Deregulation of LATS1/2, which leads to subsequent hyper-activation of YAP/TAZ, is sufficient to promote tumorigenesis in mouse models[7,8]. Furthermore, amplification of YAP and/or TAZ has been found in numerous human malignancies[9,10].

Multiple signals lead to LATS kinase activation, including contact inhibition, cellular detachment, loss of actin cytoskeletal tension, serum deprivation, glucose starvation, signaling from GPCRs, and cytokinesis failure[3,11–17]. Mechanistically, LATS kinases were initially found to be regulated by MST1/2, the mammalian orthologs of the *Drosophila* Hippo (Hpo) kinase. Activation of LATS1/2 initiates with the recruitment of MST1/2 to LATS kinases via interactions with scaffolding proteins, such as SAV1, MOB1, and NF2 at the plasma membrane[18,19]. Once recruited, MST1/2 phosphorylate LATS1/2 at their hydrophobic motifs to remove the auto-inhibitory conformations of LATS1/2, thereby allowing auto-phosphorylation and trans-phosphorylation interactions to take place at the activation loop motifs of LATS1/2. It is this phosphorylation at the activation loop that leads to full LATS kinase activity[20,21]. However, it has become increasingly clear that LATS-activating kinases are not limited to MST1/2 in mammalian cells. Genetic deletion of MST1/2 fails to prevent full LATS activation, and YAP/TAZ phosphorylation remains intact in cells lacking MST1/2[7,22]. Moreover, several conditions known to stimulate LATS activation do so in a MST1/2-independent manner, suggesting evolutionary divergence from *Drosophila* in mammalian cells, as well as the presence of additional upstream kinases that control LATS activation[7,15,17,23]. Indeed, recent work has shown the presence of additional upstream kinases controlling LATS activation outside of MST1/2, as members of the MAP4K family have been identified as having overlapping roles in directly phosphorylating the hydrophobic motif of LATS kinases[22,24]. However, cells in which *MST1/2* and all *MAP4K*s have been collectively deleted with CRISPR still induce LATS and YAP phosphorylation upon stimulation, albeit at significantly reduced levels, indicating that more upstream activators of LATS kinases exist[22,23]. Given that Hippo pathway inactivation has been found across numerous tumor types, but mutations and deletions of core Hippo signaling components are rare, identification of upstream activators of Hippo signaling carries the potential to uncover previously unappreciated tumor suppressor genes[25,26].

To identify upstream kinases that regulate LATS activity, we performed a focused RNAi screen to identify kinases contributing to LATS activity and subsequent YAP phosphorylation. This approach identified STK25 as an upstream activator of LATS kinases, whose loss significantly promotes YAP/TAZ activity. Mechanistically, we demonstrate that STK25 promotes LATS phosphorylation at the activation loop in the absence of hydrophobic motif phosphorylation, which distinguishes it from all of the other known LATS-activating kinases discovered to date.

## Results

**Identification of STK25 as a regulator of LATS signaling.** We performed a focused RNAi screen to identify kinases necessary for inducing YAP phosphorylation under conditions of activated Hippo signaling in IMR90 fibroblasts, which have previously been used to study Hippo signaling under normal, non-transformed conditions[27,28]. We focused our screen on members of the Sterile20 kinase superfamily (including MST1/2), as members of this superfamily maintain structural similarities in spite of evolutionary divergence[29]. We depleted individual kinases via RNAi and then stimulated Hippo signaling by treatment with the drug dihydrocytochalasin B (DCB), which destabilizes the actin cytoskeleton and mimics activation of Hippo signaling under loss of F-actin-driven cytoskeletal tension[30,31]. Treatment with DCB induced robust activation of Hippo signaling with consequent phosphorylation of YAP at serine 127, a LATS-specific canonical site regulating YAP cytoplasmic retention through binding to 14-3-3 proteins (Fig. 1a–c and Supplementary Fig. 1a–d)[3]. As expected, we found that depletion of known activators of LATS, such as MST1/2, decreased YAP phosphorylation, but surprisingly we found STK25 to be the strongest hit in our screen. Depletion of STK25 significantly reduced levels of YAP$^{S127}$ phosphorylation relative to controls in this assay (Supplementary Fig. 1a, b). Importantly, this effect was reproduced in multiple cell lines, including HEK293A and hTERT-RPE-1 (Fig. 1a, Supplementary Fig. 1c, d). We performed phos-tag gel electrophoresis to assess overall levels of YAP phosphorylation and found that STK25-depletion led to significantly reduced levels of phosphorylated YAP, with consequent enrichment of unphosphorylated YAP, as compared to controls (Fig. 1b). Additionally, we found that TAZ, a mammalian paralog of YAP, was also enriched in an unphosphorylated state in STK25-depleted cells (Fig. 1c). We reproduced these results in cells treated with Latrunculin A, a fungal-derived actin-binding toxin with a different mechanism of action than cytochalasin-class agents (Supplementary Fig. 1e)[32].

We used multiple approaches to ensure that decreases in YAP phosphorylation following STK25 depletion via RNAi were not due to off-target effects. First, we validated this finding with multiple distinct siRNA sequences targeting STK25 (eight total siRNAs, including three targeting the 3′UTR), and we observed that the degree of reduction in YAP phosphorylation correlated with STK25 knockdown efficiency (Supplementary Fig. 1f, g). We also confirmed that depletion of STK25 via RNAi did not negatively affect protein levels of LATS1/2 and MST1/2 in our cells (Supplementary Fig. 1i). Second, we used CRISPR-Cas9 to genetically knock out *STK25* from HEK293A and found that *STK25* KO clonal cells (generated with two different sgRNA sequences) also failed to induce YAP phosphorylation to the same extent as control cells following DCB treatment (Fig. 1d and Supplementary Fig. 1h). Finally, we demonstrated that expression of siRNA-resistant or Cas9-resistant STK25 was sufficient to rescue YAP phosphorylation in both RNAi and CRISPR-mediated depletion experiments (Fig. 1e, Supplementary Fig. 1j). By contrast, expression of kinase-dead STK25 (STK25$^{K49R}$), was not able to rescue, indicating that the observed increase in YAP phosphorylation is dependent on the kinase activity of STK25. Altogether, these data reveal that the kinase STK25 plays a previously unappreciated role in promoting YAP phosphorylation.

**STK25 depletion promotes YAP activation.** We next analyzed if the decrease in YAP phosphorylation following STK25 depletion leads to a corresponding increase in nuclear localization of active YAP. Depletion of STK25, either by RNAi or CRISPR, led to

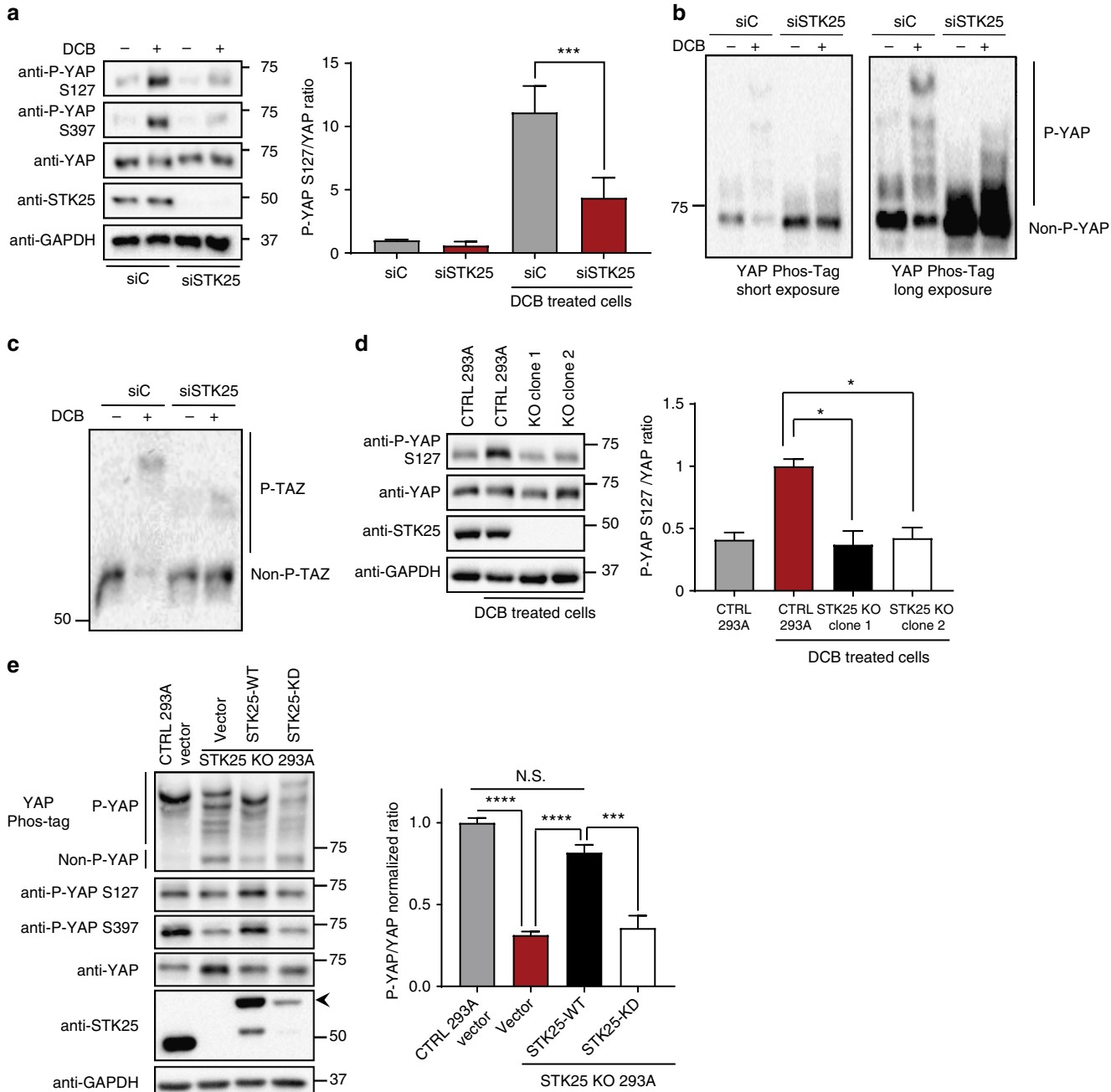

**Fig. 1** STK25 regulates Hippo activation in response to loss of cytoskeletal tension. **a** Immunoblot and quantitation of phosphorylated YAP levels following treatment with 10 μM DCB in HEK293A cells transfected with the indicated siRNA ($n = 6$; ***$p < 0.001$, unpaired $t$-test). **b** Global phosphorylation status of YAP was assessed using Phos-tag gel electrophoresis following treatment with 10 μM DCB in HEK293A cells transfected with either control siRNA or STK25 siRNA. Shifted bands indicate degrees of YAP phosphorylation. **c** TAZ phosphorylation status was assessed using Phos-tag gel electrophoresis following treatment with 10 μM DCB in HEK293A cells transfected with either control siRNA or STK25 siRNA. **d** Immunoblot and quantitation of phosphorylated YAP levels following treatment with 10 μM DCB in either control HEK293A stably expressing Cas9 and a non-targeting sgRNA or STK25 KO 293A stably expressing Cas9 together with either sgRNA 1 (Clone 1) or sgRNA 2 (Clone 2) targeting STK25 ($n = 4$; *$p < 0.05$, one-way ANOVA with Dunnett's post-hoc analysis). **e** Immunoblot and quantitation of phosphorylated YAP levels in control 293A cells and STK25 KO 293A cells transfected with either Vector, Cas9-resistant FLAG-STK25-WT, or Cas9-resistant FLAG-STK25-KD. Global levels of YAP phosphorylation in these samples were also assessed using Phos-tag gel electrophoresis. Quantitation corresponds to levels of phosphorylated YAP as measured via phos-tag electrophoresis ($n = 4$; ***$p < 0.001$, ****$p < 0.0001$; One-way ANOVA with Dunnett's post-hoc analysis). All data are presented as mean ± SEM

significant increases in nuclear YAP in multiple cell lines (Fig. 2a–c, Supplementary Fig. 2d–i). We also observed a gene dose-dependent increase in nuclear YAP in $STK25^{-/-}$ and $STK25^{+/-}$ MEFs (Supplementary Fig. 2a–c). Remarkably, we found that depletion of STK25 enabled a significant fraction of YAP to remain nuclear even under conditions of actin

depolymerization, which strongly sequesters YAP in the cytoplasm in control cells (Fig. 2d–f).

To assess whether depletion of STK25 increases YAP/TAZ activity, we first used a luciferase-based gene expression reporter assay. HEK293A were transfected with a reporter encoding a YAP/TAZ-responsive luciferase gene, in which eight

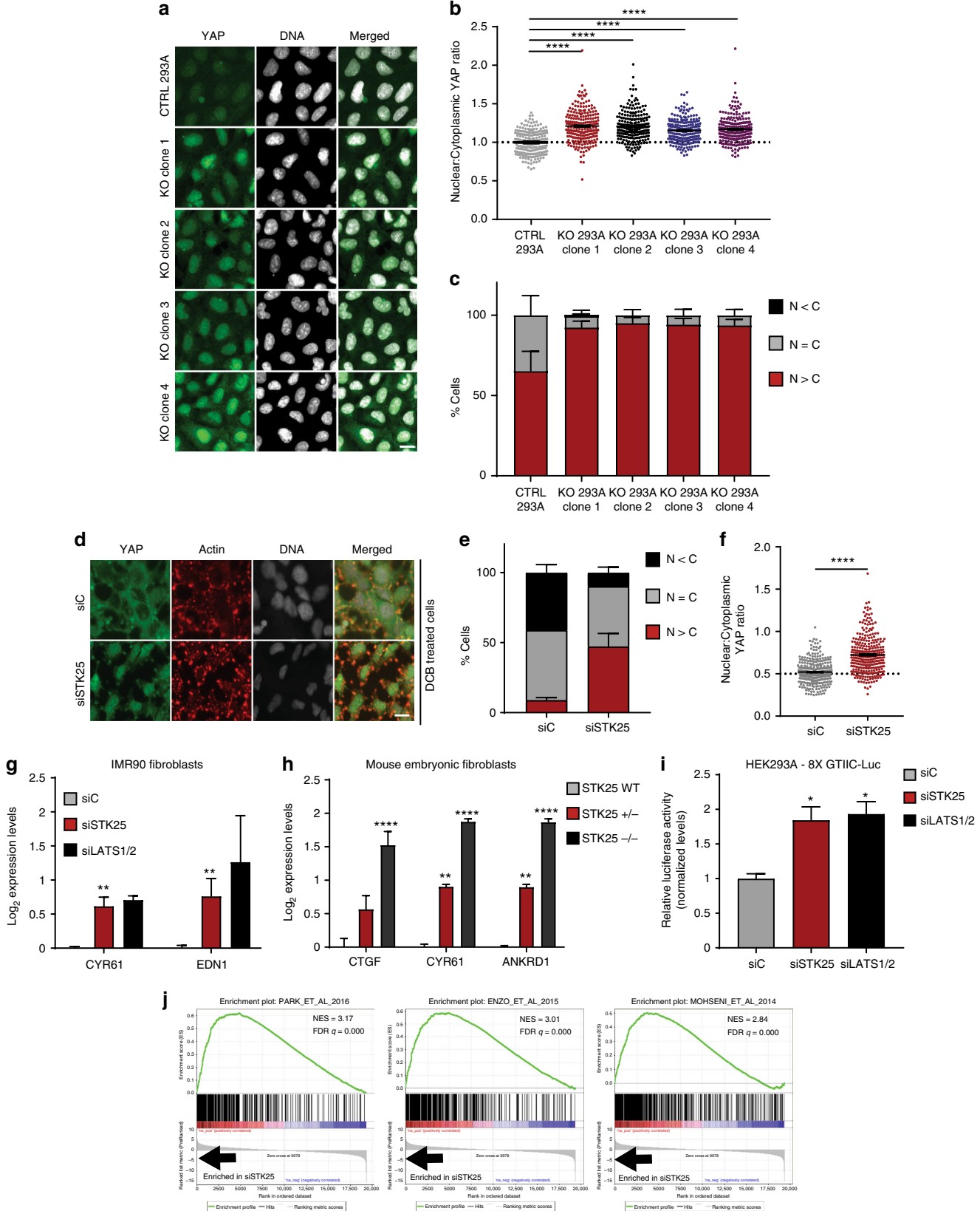

TEAD-YAP-binding sites are cloned into a promoter driving expression of firefly luciferase[11]. Using this approach, we found that STK25 depletion resulted in a doubling of expression from the luciferase reporter, indicating that loss of STK25 promotes YAP/TAZ activity (Fig. 2i). Accordingly, we discovered that

YAP-target genes were significantly upregulated in cells depleted of STK25 via RNAi or CRISPR (Fig. 2g, Supplementary Fig. 2j). We also found that YAP-target gene expression was increased in a *STK25* gene-dose-dependent fashion in *STK25*$^{+/-}$ and *STK25*$^{-/-}$ MEFs (Fig. 2h). Lastly, to assess gene expression in a

**Fig. 2** Loss of STK25 promotes activation of YAP. **a** Control and STK25 KO HEK293A were stained for YAP (green) and DNA (white). Scale bar, 20 μm. **b** Nuclear:cytoplasmic YAP ratios of control and STK25 KO HEK293A were quantified ($n = 225$ per group over three biological replicates; ****$p < 0.0001$, Kruskal–Wallis test). **c** YAP localization in control and STK25 KO HEK293A was quantified ($n = 3$ biological replicates; N>C, YAP is enriched in the nucleus; N=C, YAP is evenly distributed between the nucleus and the cytoplasm; N<C, YAP is enriched in the cytoplasm). **d** HEK293A cells transfected with either control siRNA or STK25 siRNA stained for YAP (green), Actin (red), and DNA (white) following treatment with 5 μM DCB. Scale bar, 20 μm. **e** YAP localization was quantified ($n = 3$ biological replicates; N>C, YAP is enriched in the nucleus; N=C, YAP is evenly distributed between the nucleus and the cytoplasm; N<C, YAP is enriched in the cytoplasm). **f** YAP intensity was quantified and nuclear:cytoplasmic ratios were calculated ($n = 225$ per group over three biological replicates; ****$p < 0.0001$, Mann–Whitney test). **g** qPCR analysis of YAP-target gene expression in IMR90 fibroblasts transfected with the indicated siRNA ($n = 4$; **$p < 0.01$, unpaired $t$-test). **h** qPCR analysis of YAP-target gene expression in wild-type, STK25$^{+/-}$, and STK25$^{-/-}$ mouse embryonic fibroblasts ($n = 3$ biological replicates; **$p < 0.01$, ****$p < 0.0001$, one-way ANOVA with Dunnett's post-hoc analysis). **i** Expression of the TEAD luciferase reporter in HEK293A cells transfected with the indicated siRNA. Cells were transfected with siRNA, followed by transfection with 8X GTIIC TEAD luciferase reporter and pRL-TK renilla luciferase. Reporter luciferase activity was normalized to Renilla luciferase ($n = 3$ biological replicates; *$p < 0.05$, one-way ANOVA with Dunnett's post-hoc analysis). **j** An expression signature of genes most upregulated upon loss of STK25 was constructed and GSEA was performed against a curated list of publicly available active YAP/TAZ gene sets. The top three most enriched gene sets are shown here. All data are presented as mean ± SEM

comprehensive, unbiased manner, we depleted STK25 from hTERT-RPE-1 and performed microarray analysis to obtain a list of genes that were significantly upregulated in cells lacking STK25 compared to controls. Gene set enrichment analysis (GSEA) was performed using a curated set of publicly available, published gene sets for active YAP/TAZ. This GSEA revealed that depletion of STK25 in RPE-1 cells results in a highly significant enrichment of active YAP/TAZ gene expression signatures (Fig. 2j, Supplementary Table 1)[2,33–37]. Collectively, our data demonstrate that loss of STK25 promotes YAP/TAZ activation.

**STK25 acts through LATS1/2 to inhibit YAP.** Given that loss of STK25 leads to an overall decrease in YAP phosphorylation, we predicted that overexpression of STK25 may have the opposite effect and promote YAP phosphorylation. We found that overexpression of wild-type STK25, but not a kinase dead mutant (STK25$^{K49R}$) led to significant increases in levels of phosphorylated YAP relative to controls (Supplementary Fig. 3a, b) and that this caused YAP to become enriched in the cytoplasm (Fig. 3a–c). We found that this effect was LATS1/2-dependent, as overexpression of STK25 in HEK293A genetically depleted of LATS1/2 by CRISPR (LATS1/2 dKO) did not produce cytoplasmic enrichment of YAP (Fig. 3d–f). We also depleted STK25 from both wild-type and LATS dKO 293A and found that while loss of STK25 drives YAP into the nucleus in wild-type cells, there was no such effect in LATS dKO 293A (Fig. 3g–i, Supplementary Fig. 3c), further indicating that STK25 functions through LATS1/2. Lastly, we found that knockdown of LATS1/2 via RNAi was sufficient to rescue YAP-target gene expression, even upon STK25 overexpression (Supplementary Fig. 3d). Thus, STK25 depends on LATS1/2 to regulate YAP.

Interestingly, this STK25-LATS1/2 axis is independent of other identified upstream LATS-activating kinases. HEK293A cells depleted of MST1/2 and all members of the MAP4K family (MAP4K1–7) via CRISPR-Cas9 (MM8KO 293A) still demonstrate YAP phosphorylation when grown to confluence, albeit at much lower levels than in wild-type 293A (Supplementary Fig. 3e). However, knockdown of STK25 was sufficient to reduce YAP phosphorylation even further in these MM8KO cells (Fig. 3j). Indeed, knockdown of STK25 decreased YAP phosphorylation in WT 293A and MM8KO 293A, but not LATS dKO 293A, recapitulating our finding that STK25 depends on LATS1/2 to regulate YAP (Supplementary Fig. 3e). Additionally, stable overexpression of wild-type STK25, but not kinase-dead STK25, resulted in increased levels of YAP phosphorylation in MM8KO cells (Fig. 3k). In short, our data suggest that STK25 regulates YAP in a LATS1/2-dependent fashion which is independent of the activities of MST1/2 and MAP4K family members.

**STK25 promotes LATS activation loop phosphorylation.** Given the structural similarities between STK25 and MST1/2, and our data demonstrating that STK25 is sufficient to induce YAP phosphorylation in the absence of other upstream LATS-activating kinases, we hypothesized that STK25 may phosphorylate and activate LATS. We expressed HA-tagged LATS2 and FLAG-tagged STK25 in HEK293A cells and found that HA-tagged LATS2 co-precipitated with FLAG-tagged STK25 (Fig. 4a). To assess whether LATS1 or LATS2 bind to endogenous STK25, we overexpressed either Myc-tagged LATS1 or HA-tagged LATS2, and then immunoprecipitated each respective LATS kinase. We found that endogenous STK25 co-precipitated with both LATS1 and LATS2 (Fig. 4b).

We next tested if STK25 could phosphorylate LATS. To do this, we carried out an in vitro kinase assay using HA-LATS2 as the substrate and FLAG-STK25 as the kinase following immunoprecipitation-purification from transfected cell lysates (Fig. 4c). FLAG-MAP4K1 or FLAG-MST1 served as positive controls in these reactions. As expected, we found that FLAG-MAP4K1-induced phosphorylation at the hydrophobic motif of LATS (LATS-HM), leading to subsequent phosphorylation at the activation loop (LATS-AL) (Fig. 4d). To our surprise, STK25 did not induce phosphorylation at the LATS-HM, but still induced phosphorylation at the LATS-AL (Fig. 4d). This effect was dependent on STK25 kinase activity, as a kinase-dead STK25$^{K49R}$ was unable to produce robust phosphorylation at the LATS-AL (Fig. 4d). These data suggested that STK25 might be acting to directly activate LATS through phosphorylation at the activation loop site, bypassing canonical phosphorylation at the hydrophobic motif.

To verify that STK25, and not other identified upstream activators of LATS kinases were responsible for this increase in LATS-AL phosphorylation, we utilized MM8KO 293A to IP purify STK25 and LATS2 for our in vitro kinase reactions as before. Once again, co-incubation of wild-type FLAG-STK25 with HA-LATS2 produced robust increases in phosphorylation at the activation loop of LATS2, which was decreased when LATS2 was co-incubated with kinase-dead STK25$^{K49R}$ (Supplementary Fig. 4a). As phosphorylation at the activation loop has been canonically associated with LATS auto-phosphorylation, we wished to further verify that STK25 kinase activity, and not LATS2 intrinsic kinase activity, was driving this phenomenon[20,21,38,39]. To accomplish this, we utilized LATS dKO 293A, into which kinase-dead, catalytically inactive LATS2$^{D809A}$ was transfected. FLAG-STK25 or FLAG-MAP4K1 were also transfected into LATS dKO 293A. These proteins were IP-purified as before and then used for downstream in vitro kinase assays. Such a set-up ensured that we would be utilizing

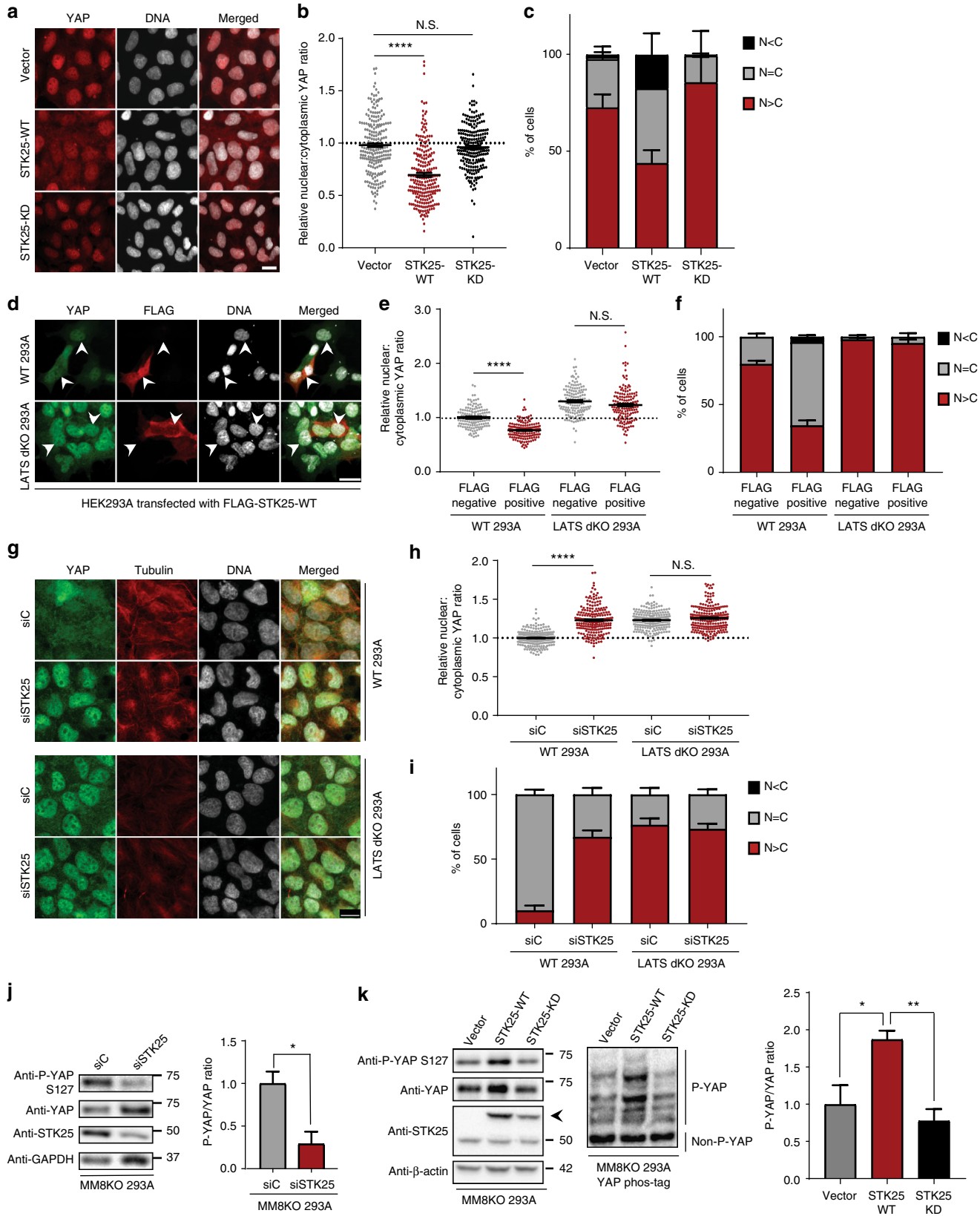

only LATS2 with no intrinsic kinase activity as our substrate, and that there would be no co-precipitating wild-type LATS1 or LATS2 to confound interpretation of the data. Our results revealed that wild-type STK25, but not kinase-dead STK25$^{K49R}$, promoted the phosphorylation of LATS-AL, independent of

LATS intrinsic kinase activity (Supplementary Fig. 4b). As expected, we found that while MAP4K1 was able to promote phosphorylation of kinase-dead LATS2 at the hydrophobic motif, it was unable to do so at the activation loop. However, we also noted the presence of LATS-AL phosphorylation even under

**Fig. 3** STK25 acts through LATS1/2 to inhibit YAP, independent of MST/MAP4Ks. **a** HEK293A cells stably expressing STK25-WT, STK25-KD, or vector were stained for YAP (red) and DNA (white). Scale bar, 20 μm. **b** YAP intensities from **a** were quantified and nuclear:cytoplasmic ratios were calculated (n = 225 per group over three biological replicates; ****p < 0.0001, Mann–Whitney test). **c** YAP localization from **a** was quantified (n = 3 biological replicates; N>C, YAP is enriched in the nucleus; N=C, YAP is evenly distributed between the nucleus and the cytoplasm; N<C, YAP is enriched in the cytoplasm). **d** Wild-type and LATS dKO HEK293A were transfected with a vector encoding FLAG-STK25-WT and were stained for YAP (green), FLAG (red), and DNA (white). Scale bar, 20 μm. Arrows indicate representative cells selected for quantification that were positive for FLAG signal (indicating expression of transfected wild-type STK25) as well as an immediately adjacent cell negative of FLAG signal also selected for quantification to serve as controls. **e** YAP intensities from **d** and nuclear:cytoplasmic ratios were calculated (n = 200 per group over four biological replicates; ****p < 0.0001, Kruskal–Wallis test with Dunn's post-test; N.S. indicates "not significant"). **f** YAP localization from **d** were quantified as before (n = 4 biological replicates). **g** Wild-type and LATS dKO HEK293A were transfected with the indicated siRNA, grown to confluence, then stained for YAP (green), Tubulin (red), and DNA (white). Scale bar, 20 μm. **h** YAP intensities from **g** were quantified and nuclear:cytoplasmic ratios were calculated (n = 225 over three biological replicates; ****p < 0.0001, Kruskal–Wallis test with Dunn's post-test; N.S. indicates "not significant.") **i** YAP localization from **g** were quantified as before (n = 3 biological replicates). **j** Immunoblot and quantification of phosphorylated YAP in MM8KO 293A cells transfected with the indicated siRNA (n = 3 biological replicates; *p < 0.05, paired t-test). **k** Immunoblot and quantitation of YAP phosphorylation in confluent MM8KO 293A cells stably expressing either the pWZL vector, STK25-WT, or STK25-KD. Quantitation corresponds to levels of phosphorylated YAP as measured via phos-tag electrophoresis (n = 4 biological replicates; *p < 0.05, **p < 0.01, one-way ANOVA with Dunnett's post-hoc analysis). All data are presented as mean ± SEM

reaction conditions involving STK25$^{K49R}$, suggesting the presence of a LATS-phosphorylating contaminant. We hypothesized that this contaminant was likely co-precipitating wild-type endogenous STK25, as STK25 has been reported to complex with itself and thus promote downstream signaling events[40]. As such, we generated a LATS1/2-STK25 triple KO (LS tKO) 293A cell line (Supplementary Fig. 4c). Generation of this LS tKO 293A cell line allowed us to IP-purify both wild-type and kinase-dead STK25 without the confounding presence of co-associated endogenous wild-type STK25, or endogenous LATS kinases (Supplementary Fig. 4c). As before, we transfected kinase-dead LATS2$^{D809A}$, FLAG-STK25, and FLAG-MST1 into LS tKO 293A, and these proteins were IP-purified and used for downstream in vitro kinase reactions. Our results demonstrated that only wild-type STK25 was able to significantly increase phosphorylation of kinase-dead LATS2 at the activation loop site (Fig. 4e). Moreover, this STK25-dependent increase in activation loop phosphorylation was not due to an increase in LATS-HM phosphorylation (Supplementary Fig. 4d). However, we do note that even in a LS tKO background, some residual LATS-AL phosphorylation, albeit non-significant, was observed upon addition of ATP in in vitro reactions, suggesting there may exist other kinases which can directly promote LATS-AL phosphorylation (Fig. 4d). In sum, these data reveal that STK25 activates LATS kinases through a mechanism that is distinct from what has been characterized for the MAP4K/MST kinases (Fig. 4d, e).

**STK25 activation of LATS does not require HM phosphorylation**. To assess whether the increase in phosphorylated LATS-AL correlates with increased LATS activity, we transfected STK25 with and without LATS2 in LATS dKO 293A and then assessed levels of YAP phosphorylation. As we previously noted, overexpression of STK25 in LATS dKO 293A was insufficient to induce YAP phosphorylation; however, transfection of wild-type LATS2 was sufficient to increase levels of phosphorylated YAP, which was further increased upon co-transfection of LATS2 with STK25$^{WT}$, but not STK25$^{K49R}$ (Supplementary Fig. 4e, f). Additionally, transfection of kinase-dead LATS2 abrogated this effect, which could not be rescued by co-transfection with STK25, indicating that STK25 depends upon LATS kinases for its inhibitory effects on YAP (Supplementary Fig. 4e, f).

To further validate our in vitro findings that STK25 promotes LATS-AL phosphorylation, we assessed the effects that modulation of STK25 has on LATS-AL phosphorylation in cells. To do this, we depleted STK25 via RNAi and stimulated LATS activity by treating cells with 10 μM DCB. We observed an increase in phosphorylated LATS-AL with DCB treatment, which was

decreased upon knockdown of STK25 (Fig. 4f). Next, we overexpressed STK25 together with LATS2 in cells and found that co-expression of LATS2 along with wild-type STK25, but not kinase-dead STK25, increased levels of phosphorylated LATS-AL (Fig. 4g).

Lastly, we assessed if STK25 is able to promote YAP phosphorylation in the absence of LATS-HM phosphorylation, based on our in vitro kinase assay results. To do this, we generated a LATS2$^{T1041A}$ mutant that cannot be phosphorylated at the hydrophobic motif site (HM-Mut-LATS2). HM-Mut-LATS2 was co-transfected into LATS dKO 293A with either STK25$^{WT}$, STK25$^{K49R}$, or MST1. This would ensure that any YAP phosphorylation observed would be solely due to the YAP-phosphorylating activity of the HM-Mut-LATS2, and not any other LATS kinase.

Strikingly, we found that transfection of HM-Mut-LATS2 alone into LATS dKO 293A was sufficient to increase YAP phosphorylation at the S127 site, suggesting that the hydrophobic motif is dispensable for LATS-dependent phosphorylation of YAP (Fig. 4h). Moreover, co-expression of HM-Mut-LATS2 and STK25$^{WT}$ was able to increase YAP$^{S127}$ phosphorylation approximately two-fold when compared to expression of HM-Mut-LATS2 alone (Fig. 4h). Further, we found that co-expression of kinase-dead STK25$^{K49R}$ with HM-Mut-LATS2 was unable to increase YAP phosphorylation levels beyond that induced by expression of HM-Mut-LATS2 alone (Fig. 4h), indicating that STK25 requires its kinase activity to increase the YAP-phosphorylating activity of HM-Mut-LATS2. Importantly, we noted that co-expression of MST1 and HM-Mut-LATS2 did not increase YAP phosphorylation relative to expression of HM-Mut-LATS2 alone, consistent with reports that MST1 requires the LATS hydrophobic motif to promote LATS activity[7,20,41]. We also noted a corresponding increase in the phosphorylation of LATS-AL upon co-expression of HM-Mut-LATS2 and STK25$^{WT}$, but not other kinases (Supplementary Fig. 4g).

Interestingly, although HM-Mut-LATS2 alone (activated by endogenous STK25) is sufficient to phosphorylate YAP, it is only half as active relative to wild-type LATS2 (activated by endogenous STK25 and MST/MAP4Ks) (Fig. 4h). As such, our data suggest that although the hydrophobic motif may be dispensable in LATS-dependent YAP phosphorylation, its presence likely serves an important function in enhancing phosphorylation of YAP by the LATS kinases, and that STK25 likely cooperates with LATS-HM kinases to regulate LATS signaling. Taken together, these data demonstrate that STK25 activates LATS by directly promoting the phosphorylation of LATS-AL, and that this mechanism appears to operate independent of LATS-HM phosphorylation and LATS intrinsic kinase activity.

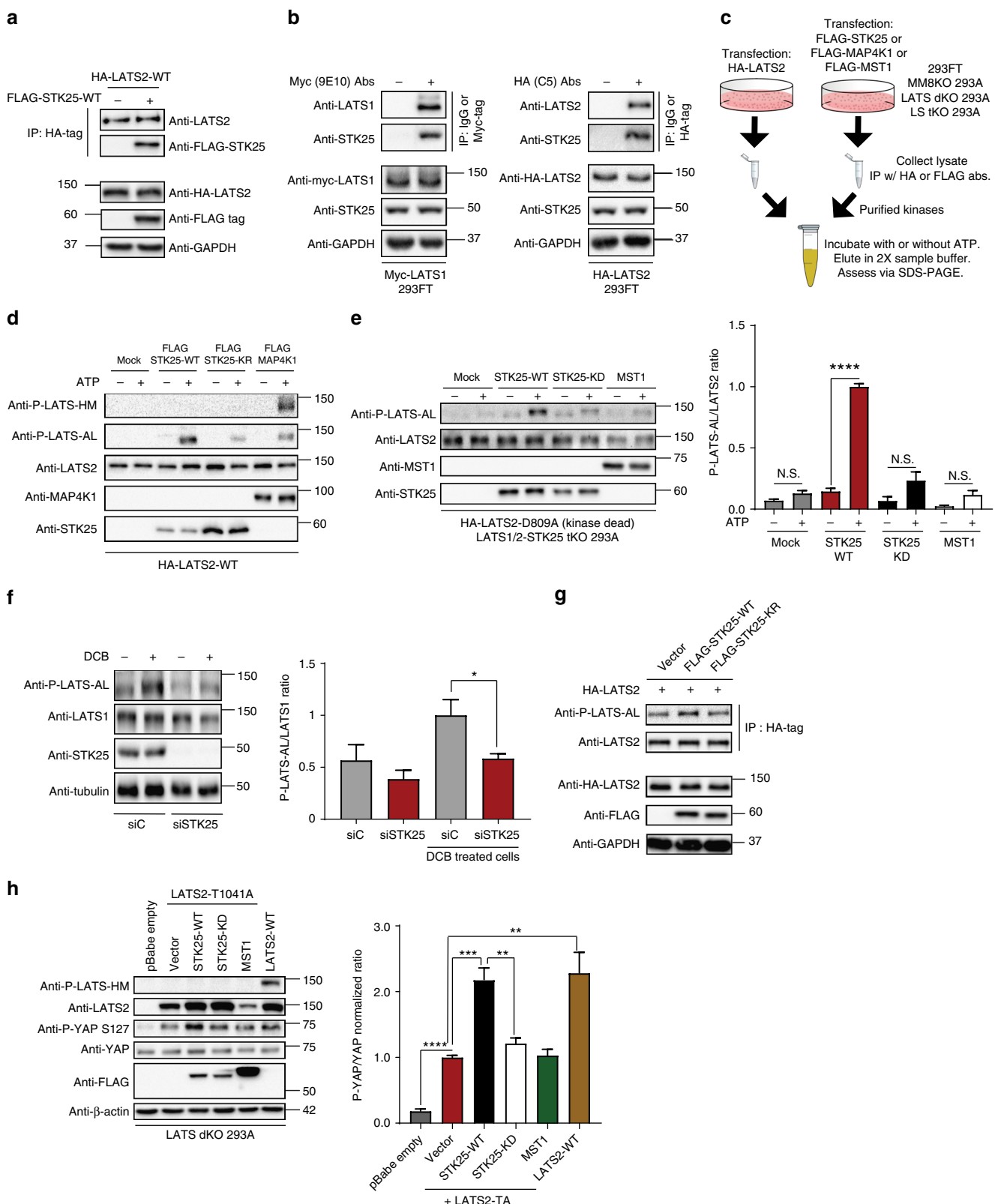

**STK25 loss impairs LATS activity under physiologic states**. We next analyzed the effects of STK25 in physiologically relevant contexts known to stimulate LATS activation. We first assessed YAP phosphorylation in confluent, contact-inhibited cells, as contact inhibition is a known LATS-activating stimulus[31]. We found that control HEK293A efficiently activated LATS and induced YAP phosphorylation upon being grown to confluence (Fig. 5a, Supplementary Fig. 5a). By contrast, HEK293A depleted of STK25 exhibited reduced levels of YAP phosphorylation (Fig. 5a, Supplementary Fig. 5b). We also observed reduced TAZ phosphorylation and subsequent stabilization of unphosphorylated TAZ levels following loss of STK25 (Supplementary

**Fig. 4** STK25 directly promotes phosphorylation of LATS activation loop. **a** LATS2 was immunoprecipitated from HEK293A cells co-transfected with HA-LATS2 and either vector control or FLAG-STK25. Co-precipitation of FLAG-STK25 with HA-LATS2 was assessed by immunoblotting. **b** HEK293A cells were transfected with Myc-LATS1 or HA-LATS2. LATS1 and LATS2 were immunoprecipitated using antibodies directed against their tags and co-precipitation of endogenous STK25 was assessed by immunoblotting. **c** Schema of the in vitro kinase assay set-up. **d** Immunoprecipitation (IP)-purified wild-type LATS2 (HA-LATS2-WT) was co-incubated with IP-purified wild-type STK25 (FLAG-STK25-WT), kinase-dead STK25 (FLAG-STK25-KD), or wild-type MAP4K1 (FLAG-MAP4K1), and assessed for phosphorylation of its hydrophobic motif (P-LATS-HM) or activation loop (P-LATS-AL). **e** IP-purified kinase-dead LATS2 (HA-LATS2-KD) from transfected LATS1/2-STK25 triple KO 293A cells (LS tKO 293A) was co-incubated with IP-purified FLAG-STK25-WT, FLAG-STK25-KD, or FLAG-MST1, all from transfected LS tKO 293A. Levels of phosphorylated LATS at the activation loop (P-LATS-AL) were assessed via immunoblotting. Levels of phosphorylated LATS2-KD at the activation loop were then quantitated via densitometry ($n = 4$ biological replicates; ****$p < 0.0001$, N.S. indicates "not significant," unpaired $t$-test). **f** Immunoblot and quantitation of LATS activation loop phosphorylation (P-LATS-AL) following treatment with 10 μM DCB in HEK293A cells transfected with the indicated siRNA ($n = 4$ biological replicates; *$p < 0.05$, paired $t$-test). **g** HEK293A cells were co-transfected with HA-tagged wild-type LATS2 (HA-LATS2-WT) and either vector control (Vector), wild-type STK25 (FLAG-STK25-WT), or kinase-dead STK25 (FLAG-STK25-KD). LATS2 was immunoprecipitated and used to assess levels of activation loop phosphorylation by immunoblotting. Input lysates were assessed by immunoblotting for assessing protein loading and verification of transfected protein expression. **h** Immunoblot and quantitation of YAP phosphorylation in LATS dKO 293A cells transfected with the indicated expression plasmids. LATS2-TA indicates hydrophobic motif mutant LATS2 $^{T1041A}$ ($n = 4$ biological replicates; **$p < 0.01$, ***$p < 0.001$, ****$p < 0.0001$, one-way ANOVA with Dunnett's post-hoc analysis). All data are presented as mean ± SEM

Fig. 5c). These results were reproduced in STK25 KO 293A (Supplementary Fig. 5d). We also grew adherent cells in suspension for defined periods of time, as cell detachment is another known LATS-activating condition[31]. We found that depletion of STK25 reproducibly impaired phosphorylation of YAP under conditions of cell detachment (Fig. 5c). Lastly, we grew to confluence either control or STK25 KO 293A cells, and assessed levels of phosphorylated LATS1 at the hydrophobic motif (LATS-HM) and the activation loop (LATS-AL). We found a significant decrease in phosphorylated LATS-AL in our STK25 KO 293A, but to our surprise, we also noted a robust and seemingly compensatory increase in phosphorylation of LATS-HM (Fig. 5g). This data suggested that cells lacking STK25 are unable to appropriately phosphorylate LATS-AL even when the LATS-HM is phosphorylated (Fig. 5g). Together, these results demonstrate that STK25 plays a significant role in activating LATS kinases following physiologically relevant cellular perturbations known to activate Hippo signaling.

We hypothesized that depletion of STK25, with subsequent YAP activation, would also provide proliferative advantages to cells. Indeed, we found that STK25 KO 293A have increased growth rates in culture compared to controls (Fig. 5b). We also found that STK25 depletion allows cells to partially overcome cell cycle arrest induced by contact inhibition, thereby validating our finding that STK25 loss prevents YAP phosphorylation under contact inhibited conditions (Fig. 5d). Further, it has been demonstrated that tetraploid cells arising from cytokinetic failures fail to proliferate efficiently due to LATS activation and subsequent YAP inhibition. Proliferation can be restored to tetraploid cells through restoration of YAP activity[15]. Correspondingly, we found that STK25 knockdown is also sufficient to restore proliferative capacity to tetraploid cells (Fig. 5e, f).

**STK25 regulates LATS-YAP activity in vivo.** We next sought to assess whether STK25 regulates LATS activity in vivo. Several studies have demonstrated that decreased Hippo signaling in mice promotes liver overgrowth. Thus, we aged both $STK25^{+/+}$ and $STK25^{-/-}$ mice to 6–12 months and assessed liver mass. We found that $STK25^{-/-}$ mice had a moderate, but significant, increase in liver mass, consistent with other transgenic mouse models of inactive Hippo signaling, such as knockouts of $Nf2$ or $Sav1$ (Fig. 6a, b)[18,42]. Interestingly, one $STK25^{-/-}$ mouse presented with severe hepatomegaly at ~11 months of age (Supplementary Fig. 6a). Analysis of RNA extracts from these mouse livers revealed increased YAP-target gene expression in the $STK25^{-/-}$ mouse livers as compared to their wild-type controls (Fig. 6c). Protein

samples were also extracted from these livers, which were then analyzed for Hippo signaling status (Fig. 6d). We found that, as with our cell culture models, $STK25$ loss resulted in decreased LATS-AL phosphorylation, but not LATS-HM phosphorylation (Fig. 6d, e). Moreover, we noted a decrease in YAP phosphorylation at S397, a canonical LATS-dependent site, but not at S127, which is known to be under control by MST1/2 in the mouse liver (Fig. 6d, e)[7]. Importantly, we noted an increase in the total levels of YAP and TAZ in the $STK25^{-/-}$ livers, which was confirmed in stained sections of livers taken from both $STK25^{+/+}$ and $STK25^{-/-}$ mice (Fig. 6g, h, Supplementary Fig. 6b). We also assessed levels of PCNA, a well-established marker of cell proliferation, and found that $STK25^{-/-}$ livers had significantly increased levels of PCNA protein as compared to wild-type livers, consistent with our findings that $STK25$ loss promotes liver overgrowth (Fig. 6d, f). Altogether, our data indicates that loss of $STK25$ decreases Hippo signaling under physiologic conditions in vivo.

**STK25 is a putative tumor suppressor gene.** Our data demonstrate that STK25 is an activator of LATS kinases whose depletion leads to activation of YAP/TAZ and subsequent cellular proliferation. YAP/TAZ are known to be hyper-activated across a spectrum of human malignancies, although mechanisms leading to their activation remain poorly understood. Our data suggest that loss of STK25 may represent one route through which cancer cells functionally activate YAP/TAZ. Interestingly, bioinformatic analysis of TCGA reveals that focal deletion of $STK25$ is a common event across many tumor subtypes, with deep deletions occurring in a significant proportion of multiple aggressive cancers (Table 1, Fig. 7a). For example, $STK25$ is homozygously deleted in nearly 9% of all sarcomas (Table 1, Fig. 7a). We assessed if $STK25$ deletion status has an effect on the clinical course of disease and found that sarcoma patients with $STK25$ deletions have significantly shorter durations of survival and that deletions of $STK25$ are more common in patients with recurrent disease than those without (Fig. 7c, Supplementary Fig. 7a). Additionally, we found that this pattern of recurrent focal loss in human cancers is unique to $STK25$ among the currently identified LATS-activating kinases (Supplementary Table 2). Moreover, we noted that while the $STK25$ focally deleted peak contained 18 additional genes, none of them appeared to be known tumor suppressor genes (Supplementary Table 3). Taken together, our findings suggest that loss of $STK25$ may be one mechanism through which human cancers functionally inactivate Hippo signaling to promote tumorigenesis and disease progression.

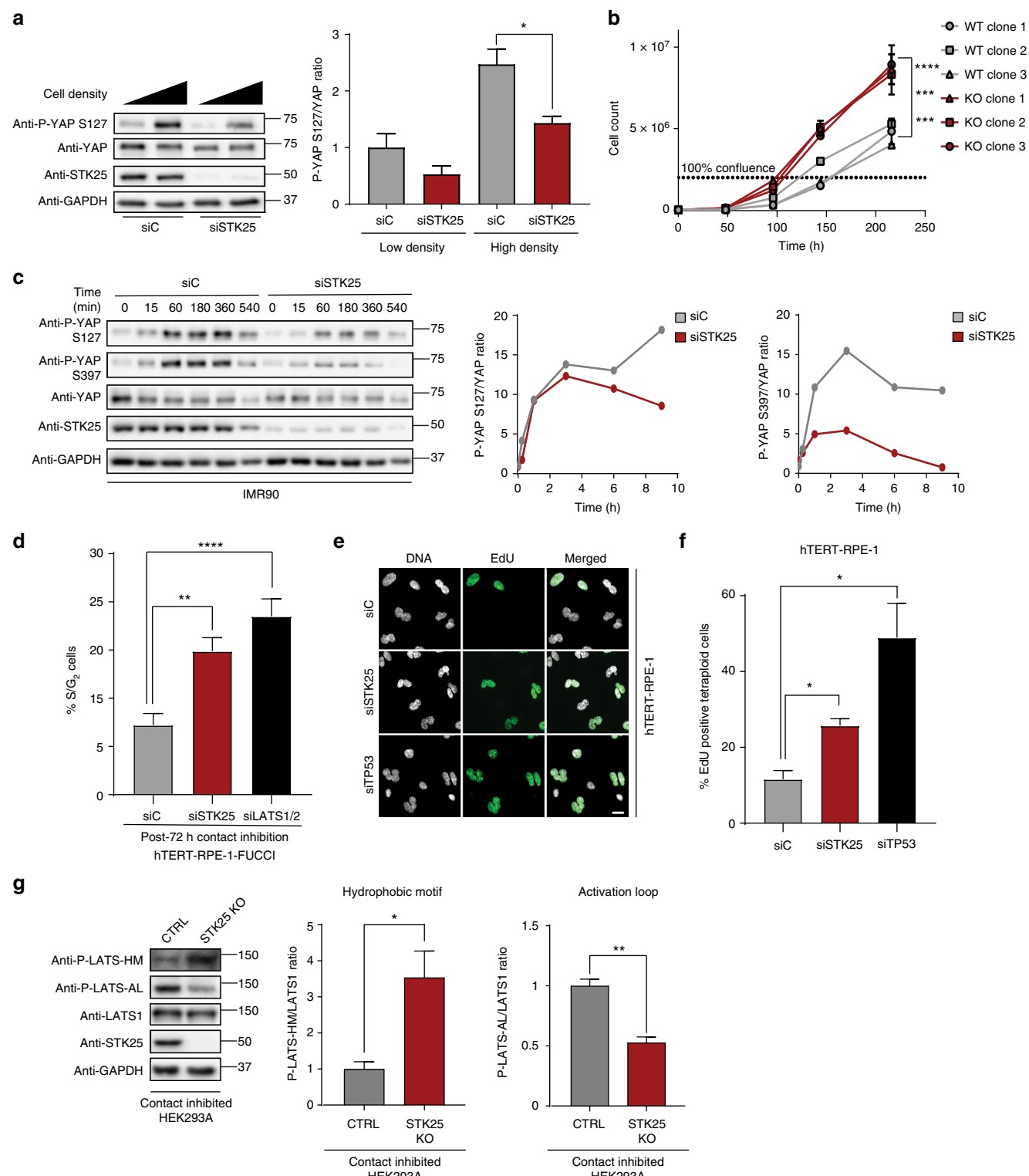

## Discussion

Although MST1/2 are known activators of LATS1/2, it has been established that genetic ablation of both *MST1/2* does not completely inactivate LATS signaling in mammalian cells[7,22], indicating the presence of additional LATS regulators[29]. Indeed, members of the MAP4K family were recently identified as upstream activators of LATS kinases, wherein MAP4Ks promote LATS activation via phosphorylation of the LATS hydrophobic motif[21,22,24,43]. Here, we identify the kinase STK25 as an additional activator of LATS kinases. However, unlike all other LATS-activating kinases discovered to date, we demonstrate that STK25 promotes phosphorylation of the activation loop motif of LATS, independent of a preceding hydrophobic motif phosphorylation.

This alternative mechanism provides some explanation for the robustness with which loss of STK25 is able to decrease LATS activation, and may also explain why *STK25* is frequently focally

**Fig. 5** STK25 regulates YAP phosphorylation in response to physiologic stimuli. **a** Immunoblot and quantification of YAP phosphorylation in HEK293A cells grown to low confluence or high confluence after transfection with the indicated siRNA ($n = 3$ biological replicates; *$p < 0.05$, unpaired $t$-test). **b** Cellular proliferation curves of control HEK293A clones and STK25 KO clones over the indicated time periods ($n = 3$ replicates per cell line; ***$p < 0.001$, ****$p < 0.0001$, two-way ANOVA with Tukey's post-hoc test). **c** Representative immunoblot and quantification of YAP phosphorylation in IMR90 fibroblasts transfected with the indicated siRNA and held in suspension for the indicated time periods. **d** Quantification of the percentage of cells remaining in S/G$_2$ phase following prolonged contact inhibition in hTERT-RPE-1-FUCCI cells transfected with the indicated siRNA ($n = 4$ biological replicates; **$p < 0.01$, ****$p < 0.0001$, one-way ANOVA with Dunnett's post-hoc test). **e** Cytokinesis failure was pharmacologically induced in hTERT-RPE-1 cells to generate binucleated tetraploid cells, and the percentage of EdU-positive tetraploid cells following siRNA transfection was quantified. Cells were stained for DNA (white) and EdU incorporation (green). Scale bar, 20 μm. **f** Quantification of the percentage of EdU-positive binucleated tetraploid cells following transfection with the indicated siRNA. TP53 siRNA served as positive control ($n = 4$ biological replicates; *$p < 0.05$, one-way ANOVA with Dunnett's post-hoc test). **g** Immunoblot and quantification of LATS1 hydrophobic motif (P-LATS-HM) and activation loop (P-LATS-AL) phosphorylation in either control HEK293A stably expressing Cas9 and a non-targeting sgRNA or STK25 KO 293A stably expressing Cas9 together with sgRNA 1 grown to confluence ($n = 3$ biological replicates; *$p < 0.05$, **$p < 0.01$, paired $t$-test). All data are presented as mean ± SEM

deleted in a spectrum of human cancers, while *MST/MAP4K* pathway components are not (Supplementary Table 2). Indeed, it has previously been demonstrated that while genetic deletion of all MST/MAP4K pathway components potently decreases LATS and YAP phosphorylation, it is still possible to induce LATS activity upon treatment with actin depolymerizing agents or contact inhibition[22–24,44]. We have now shown that STK25 is responsible for at least a portion of this remaining LATS activity. Additionally, we demonstrate that deletion of *STK25* results in decreased LATS activity and increased YAP/TAZ activity in vivo. While *STK25* KO mice are viable, they exhibit significantly enlarged livers, consistent with other mouse models in which upstream LATS regulators, such as *Nf2* or *MST1/2*, are deleted (Fig. 6d, e)[7,18]. It is important to note that *STK25* deletion in vivo does not abolish LATS signaling, and therefore does not pheno-copy complete loss of *LATS1/2*, which results in non-functional livers due to abnormal differentiation of hepatocytes into biliary epithelial lineage cells and embryonic lethality[45].

Several questions still remain regarding regulation of Hippo signaling, especially with respect to how stimulatory inputs that induce loss of cytoskeletal tension ultimately activate kinases upstream of LATS[7,22,24]. Interestingly, TAO kinases have been shown to promote LATS activity by directly phosphorylating LATS-HM and by activating MST/MAP4Ks[23,46–48], suggesting STK25 may be subject to similar regulatory control. Indeed, a recent study has suggested that TAO kinases may act upstream of GCKIII (comprising MST3, MST4, and STK25 in mammals) to regulate tracheal morphogenesis in Drosophila[49]. Alternatively, STK25 may be constitutively active and loss of cytoskeletal tension simply promotes LATS–STK25 interaction. It was recently shown that TRIP6 negatively regulates LATS by competing for MOB1 binding, and that this is relieved upon loss of cytoskeletal tension, suggesting that loss of tension allows for recruitment of LATS to other binding partners that promote its activity[17]. Another possibility involves spatial regulation of STK25 in tandem with the status of the Golgi apparatus. STK25 is known to localize to and regulate the polarization of the Golgi ribbon[50,51]. Recent studies have implicated the Golgi as serving a sensor role that integrates extracellular signals for pathways regulating cellular proliferation[52]. It is possible that when the Golgi becomes disorganized under conditions of low cytoskeletal tension, such as when actin becomes pharmacologically disrupted[53], that STK25 becomes released from the Golgi to associate with LATS kinases. Alternatively, loss of tension and consequent Golgi disruption may serve as a signal to recruit LATS kinases to sites of disturbed Golgi, where they can then associate with STK25 to promote LATS activity.

In conclusion, we define STK25 as a regulator of Hippo signaling, which activates LATS1/2 via a mechanism independent from the canonical MST/MAP4K-signaling pathway, thereby explaining why YAP/TAZ activation secondary to STK25 loss cannot be completely compensated for by MST/MAP4Ks. We posit that *STK25* is a putative tumor suppressor gene, with data from human cancers supporting our claim: *STK25* undergoes significant focal deletions in a large variety of human cancers, and loss of STK25 in our cellular models promotes increased proliferation and resistance to stimuli that would normally induce cell cycle arrest. Deletions or mutations of core Hippo pathway components are rare, and it remains to be discovered how cancer cells overcome this critical tumor suppressor pathway during neoplastic transformation. Our data demonstrate that loss of STK25 represents one potential route through which cancer cells might deregulate the Hippo pathway to achieve pathologic capacity.

## Methods

**Cell culture**. HEK293A cells were obtained from Invitrogen. 293FT, hTERT-RPE-1, and IMR90 fibroblasts were purchased from ATCC. HEK293A and IMR90 were cultured in high glucose DMEM (Gibco) supplemented with 10% fetal bovine serum (ThermoFisher) with 100 IU/mL penicillin, 100 μg/mL streptomycin (ThermoFisher), and 5 μg/mL plasmocin (InvivoGen). For all experiments involving IMR90 fibroblasts, cells between passages 11 and 17 were used. 293FT and hTERT-RPE-1 were cultured in DME/F12 supplemented with 10% FBS and 100 IU/mL penicillin, 100 μg/mL streptomycin, and 5 μg/mL plasmocin (Invivo-Gen). STK25 transgenic MEFs were provided by Dr. Brian Howell of SUNY Upstate, and were cultured in high glucose DMEM supplemented with 10% FBS, 100 IU/mL penicillin, 100 μg/mL streptomycin, 5 μg/mL plasmocin (InvivoGen), 1X concentration of non-essential amino acids (Gibco), and 1 mM sodium pyruvate (Gibco). For all experiments involving MEFs, cells between passages 2 and 5 were used, and all samples were appropriately passage matched between different genotypes. Wild-type, MM8KO, and LATS dKO HEK293A were generous gifts from Kun-Liang Guan, and were cultured in high glucose DMEM (Gibco) supplemented with 10% fetal bovine serum (ThermoFisher) with 100 IU/mL penicillin, 100 μg/mL streptomycin (ThermoFisher), and 5 μg/mL plasmocin (InvivoGen). All cells were maintained at 37 °C with a 5% CO$_2$ atmosphere.

**Generation of STK25 transgenic mouse embryonic fibroblasts**. Pregnant *STK25$^{+/+}$*, *STK25$^{+/-}$*, and *STK25$^{-/-}$* female mice were euthanized at ~2 weeks post-appearance of the copulation plug and embryos were extracted via dissection. Embryos were rinsed once with Hank's balanced salt solution (HBSS), and had placental and maternal tissues removed mechanically. At this point, the heads and red organs were dissected away from the embryos and then rinsed twice with ice-cold HBSS. The embryo bodies were then trypsinized following mechanical mincing with scalpel blades. The tissue–trypsin mixture was then thoroughly mixed via pipetting and incubated at 37 °C for 20–30 min. Trypsin was then inactivated by the addition of 500 μL of FBS and the mixture was centrifuged at 168×g for 3 min. The supernatant was removed, and DNase I was added, at which point cells were further dissociated via pipetting. The dissociated cells were then passed through a 70 μm cell strainer and re-centrifuged. The cell pellet was then re-suspended in standard MEF media (high glucose DMEM supplemented with 10% FBS, 100 IU/mL penicillin, and 100 μg/mL streptomycin, 1X concentration of non-essential amino acids, and 1 mM sodium pyruvate). The following morning, the old media was removed and fresh MEF media was added to cells. Once cells reached 70–80% confluence, they were harvested via trypsinization and frozen as a suspension in cell freezing media (MEF media supplemented with 10% DMSO) until further use.

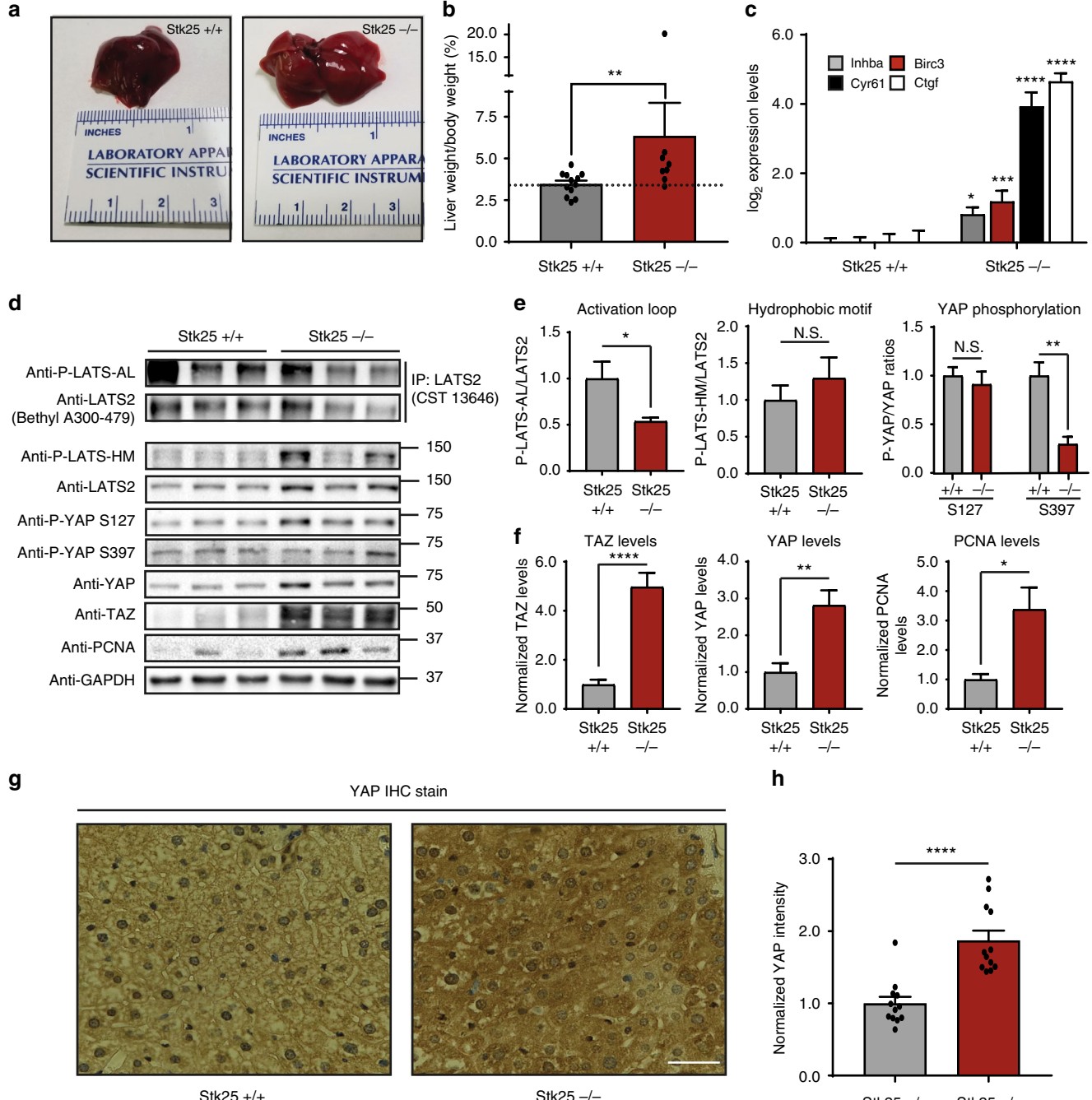

**Fig. 6** Loss of *STK25* inactivates Hippo signaling in vivo. **a** Representative photographic images demonstrating gross morphology and size of livers dissected from *STK25*$^{+/+}$ or *STK25*$^{-/-}$ mice. **b** Livers from *STK25*$^{+/+}$ and *STK25*$^{-/-}$ mice were dissected and weighed; liver/body weight ratios were then plotted for analysis ($n = 12$ mice for *STK25*$^{+/+}$ mice; $n = 8$ for *STK25*$^{-/-}$ mice. **$p < 0.01$, Mann–Whitney test). **c** qPCR analysis of validated YAP-target genes in the livers of *STK25*$^{+/+}$ and *STK25*$^{-/-}$ mice ($n = 3$ biological replicates; *$p < 0.05$, ***$p < 0.001$, ****$p < 0.0001$, unpaired *t*-test). **d** Representative immunoblot of Hippo signaling components in the livers of *STK25*$^{+/+}$ and *STK25*$^{-/-}$ mice. To probe for P-LATS-AL, endogenous LATS2 was first immunoprecipitated from tissue lysates and then re-analyzed via SDS–PAGE for phosphorylation status. **e** Quantitation of LATS phosphorylation and YAP phosphorylation in the livers of *STK25*$^{+/+}$ and *STK25*$^{-/-}$ mice from **d** ($n = 4$ biological replicates, *$p < 0.05$, **$p < 0.01$, N.S. indicates "not significant," unpaired *t*-test). **f** Quantitation of total protein levels of interest in the livers of *STK25*$^{+/+}$ and *STK25*$^{-/-}$ mice ($n =$ at least four biological replicates, *$p < 0.05$, **$p < 0.01$, ****$p < 0.0001$, unpaired *t*-test). **g** IHC staining for YAP was performed on sections of livers from *STK25*$^{+/+}$ and *STK25*$^{-/-}$ mice. Representative ×40 images are presented here. Scale bar, 100 μm. **h** Quantitation of YAP staining intensity from **g** was performed and plotted for analysis ($n = 4$ biological replicates, each with three randomly chosen fields of view for quantitation; ****$p < 0.0001$, Mann–Whitney test). All data are presented as mean ± SEM

### Table 1 Focal deletion of *STK25* is common in human cancers

| Cancer type | Q-value | Frequencies | | |
| --- | --- | --- | --- | --- |
| | | Overall | Focal deletion | High-level deletion |
| Cervical squamous cell carcinoma | 5.99E−43 | 0.4034 | 0.2814 | 0.0068 |
| Sarcoma | 2.38E−35 | 0.4297 | 0.2734 | 0.0898 |
| Bladder urothelial carcinoma | 6.81E−33 | 0.4297 | 0.2647 | 0.0172 |
| HNSCC | 2.08E−26 | 0.4951 | 0.1935 | 0.0000 |
| Kidney cancers | 1.05E−25 | 0.2989 | 0.0578 | 0.0023 |
| Brain lower grade glioma | 6.16E−24 | 0.1259 | 0.1014 | 0.0039 |
| Lung cancers | 1.82E−22 | 0.1481 | 0.1396 | 0.0000 |
| Lung squamous cell carcinoma | 2.71E−21 | 0.2183 | 0.2136 | 0.0000 |
| Kidney RCC carcinoma | 1.93E−19 | 0.0947 | 0.0701 | 0.0019 |
| Glial cancers | 4.42E−17 | 0.1239 | 0.0670 | 0.0018 |
| Breast invasive adenocarcinoma | 5.75E−16 | 0.2944 | 0.1157 | 0.0019 |
| Ovarian serous cystadenocarcinoma | 3.19E−15 | 0.3040 | 0.2228 | 0.0121 |
| Stomach adenocarcinoma | 4.89E−6 | 0.1633 | 0.1043 | 0.0000 |
| **All Cancers** | **5.6E−223** | **0.1892** | **0.1050** | **0.0049** |

Publicly available TCGA datasets were probed to assess rates of focal deletion in STK25 using the Tumorscape program online (http://www.broadinstitute.org/tcga/). The top 10 cancers with the highest rates of focal deletions in STK25 are shown, together with the "All Cancers" dataset

**Cell culture-dependent Hippo signaling analysis**. IMR90 fibroblasts were transfected with siRNA pools directed against individual kinase members of the Sterile20 superfamily of kinases. Forty-eight hours post-transfection, these fibroblasts were treated with 10 μM dihydrocytochalasin B (DCB) for 1 h to acutely induce loss of actin cytoskeletal tension and thereby activate Hippo signaling. Protein lysates were then collected and levels of YAP phosphorylation were assessed via quantitative immunoblotting. A total of three independent replicates were performed to obtain a list of kinases which reduced YAP phosphorylation under DCB treatment in these cells. The top four hits were then knocked down via siRNA transfection again in hTERT-RPE-1 cells, and DCB treatment was performed again as before. Quantitative immunoblotting was performed on RPE-1 lysates to assess YAP phosphorylation; a total of four independent replicates were performed. To further assess the effects of STK25 loss in HEK293A cells, Hippo signaling was activated by treatment with either 10 μM DCB or 1 μg/mL Latrunculin A for 1 h to acutely disrupt the actin cytoskeleton, after which proteins were collected for downstream immunoblotting applications. To prevent other Hippo activating stimuli under conditions of pharmacologic actin disruption, cells were plated so that they would be sub-confluent at the time of DCB or Latrunculin A treatments. To activate Hippo signaling physiologically, cells were either grown to confluence, or were trypsinized and placed into cell culture media in a conical tube and incubated in suspension with end-over-end rocking to prevent adhesion. Protein samples were collected as before for immunoblotting applications. DCB was obtained from Sigma-Aldrich. Latrunculin A was obtained from Tocris Bioscience.

**Generation of plasmids and stable cell lines**. Retroviral pWZL Blast Myc vector was a gift from William Hahn (Addgene, 10674) and bacterial expression vector pLDNT7 NFLAG-STK25 was obtained from DNASU (HsCD00298674). Both vectors were digested with BamHI and XhoI restriction enzymes (New England Biolabs) and fragments of interest were agarose gel purified. The NFLAG-STK25 insert was ligated into linearized pWZL Blast vector using T4 DNA ligase and transformed into DH5-Alpha chemically competent *Escherichia coli*. Colonies were screened for inserts of correct size via restriction enzyme digest and the presence of gene of interest was verified by Sanger sequencing. To generate the kinase-dead STK25 expression vector, Lysine 49 of STK25 was mutated to an Arginine using cDNA sequence-specific primers and Q5 Site-Directed Mutagenesis kit from New England Biolabs, and the presence of the correct mutation, as well as the absence of other mutations were confirmed via Sanger sequencing. For generation of STK25-targeting all-in-one CRISPR lentiviral vectors to concurrently express Cas9 and gRNA, two sgRNA targeting STK25 (sgRNA 1: 5′- TGGATCATCATGGAGTAC CTGGG-3′; sgRNA 2: 5′-TATGTCTCCTCCAGGGGACCTGG-3′) were inserted into lentiCRISPR v2 (Addgene, 52961). A lentiCRISPR v2 vector with a non-targeting sgRNA sequence served as control. To generate Cas9-resistant STK25 expression vectors, the region of STK25 cDNA corresponding to sgRNA 1-binding site were identified (5′-C AAG CTA TGG ATC ATC ATG GAG TAC CTG GGC-3′) and five consecutive silent mutations were introduced into the region, including the PAM motif to generate a Cas9-resistant sequence (5′-C AAG CTA TGG *ATA ATA* ATG *GAA TAT CTA* GGC-3′) in both pWZL Blast FLAG-STK25-WT and pWZL Blast FLAG-STK25-KD using mutation-specific primers and the Q5 site-directed mutagenesis kit (New England Biolabs). Clones were screened via restriction enzyme digests, the ability to induce expression of STK25 in STK25 KO 293A cells, and Sanger sequencing. To generate viral particles, 293FT cells were transiently transfected with either retroviral or lentiviral vectors encoding genes of interest alongside appropriate packaging plasmids, after which cell culture supernatant was collected via centrifugation and sterile filtration. Viral infections were performed by incubating cells with viral supernatant alongside 10 μg/mL polybrene (Santa Cruz Biotechnology) for 16 h, after which cells were allowed to recover for 24 h prior to application of antibiotic selection with either 5 μg/mL of Blasticidin (Sigma-Aldrich) or 4 μg/mL of Puromycin (Santa Cruz Biotechnology), depending on the viral vector being used. For overexpression experiments, pools of stably infected cells were collected and used, while further single cell cloning with cloning cylinders (Fisher Scientific) were performed for experiments involving CRISPR-Cas9 knockout cell lines.

**Plasmids and transfections**. pWZL Blast GFP was from Robert Weinberg (Addgene, 12269). pWZL Blast Myc was from William Hahn (Addgene, 10674). pcDNA3.1 FLAG-MAP4K1 was a generous gift from Duojia Pan. pcDNA3.1 FLAG-HA was from Adam Antebi (Addgene 52535). pEXPR 3F-MST1 was a generous gift from Xaralabos Varelas. LATS vectors were generated using PCR-based cloning into pBabe retroviral vectors. For siRNA transfections, 50 nM of siRNA were transfected using Lipofectamine RNAiMAX (ThermoFisher Invitrogen) according to the manufacturer's instructions. For DNA transfections, plasmid DNA were transfected using Lipofectamine 3000 (ThermoFisher Invitrogen) according to the manufacturer's instructions. Cells were harvested for downstream applications 48–72 h post-transfection. For the CRISPR rescue experiments, $7.5 \times 10^5$ and $1.5 \times 10^6$ cells were plated onto poly-D-lysine (Millipore) coated six-well tissue culture plates in serum-free, antibiotic-free DMEM, taking care to compensate for differences in cell growth rates. After allowing cells to adhere overnight, cells were transfected at a ratio of 2 μg of plasmid DNA per $1.5 \times 10^6$ cells using Lipofectamine 3000 (ThermoFisher Invitrogen) according to the manufacturer's instructions for 4 h. Cell culture media was then changed to high glucose DMEM supplemented with 10% FBS, 100 IU/mL penicillin and 100 μg/mL streptomycin, and cells were allowed to grow for 36–48 h, at which point cell lysates were collected for downstream assessments. For transfections of LATS dKO and LS tKO 293A cells for the in vitro kinase assays, cells were plated to a final density of ~60% on the day of transfection in serum-free, antibiotic-free DMEM and allowed to adhere overnight. On the following morning, cells were transfected with 14 μg of plasmid DNA with Lipofectamine 3000 (ThermoFisher Invitrogen) according to the manufacturer's instructions. Protein lysates were collected 48–72 h post-transfection for downstream use. Please refer to Supplementary Table 4 for a list of all siRNA oligonucleotide sequences used for this study.

**Synthetic YAP/TAZ Luciferase reporter assay**. HEK293A cells were plated in technical triplicates for each treatment group in a 96-well plate and transfected with siRNAs of interest as before. Twenty-four hours post-siRNA transfection, cells were transfected with both the 8X GTIIC luciferase reporter vector and the pRL-TK renilla luciferase vector using FuGENE HD transfection reagent (Promega) according to manufacturer's instructions. 8X GTIIC luciferase was a generous gift from Bob Varelas. pRL-TK was obtained from Promega (E2241). Twenty-four hours post-DNA transfection, cells were lysed in passive lysis buffer and luciferase activity levels were assessed using the Pierce Firefly Luciferase Glow and the Pierce Renilla Luciferase Glow kits according to manufacturer's instructions. Firefly luciferase activity levels were normalized to levels of Renilla luciferase activity in each well. A total of three independent experiments were performed and results are reported as mean ± SEM.

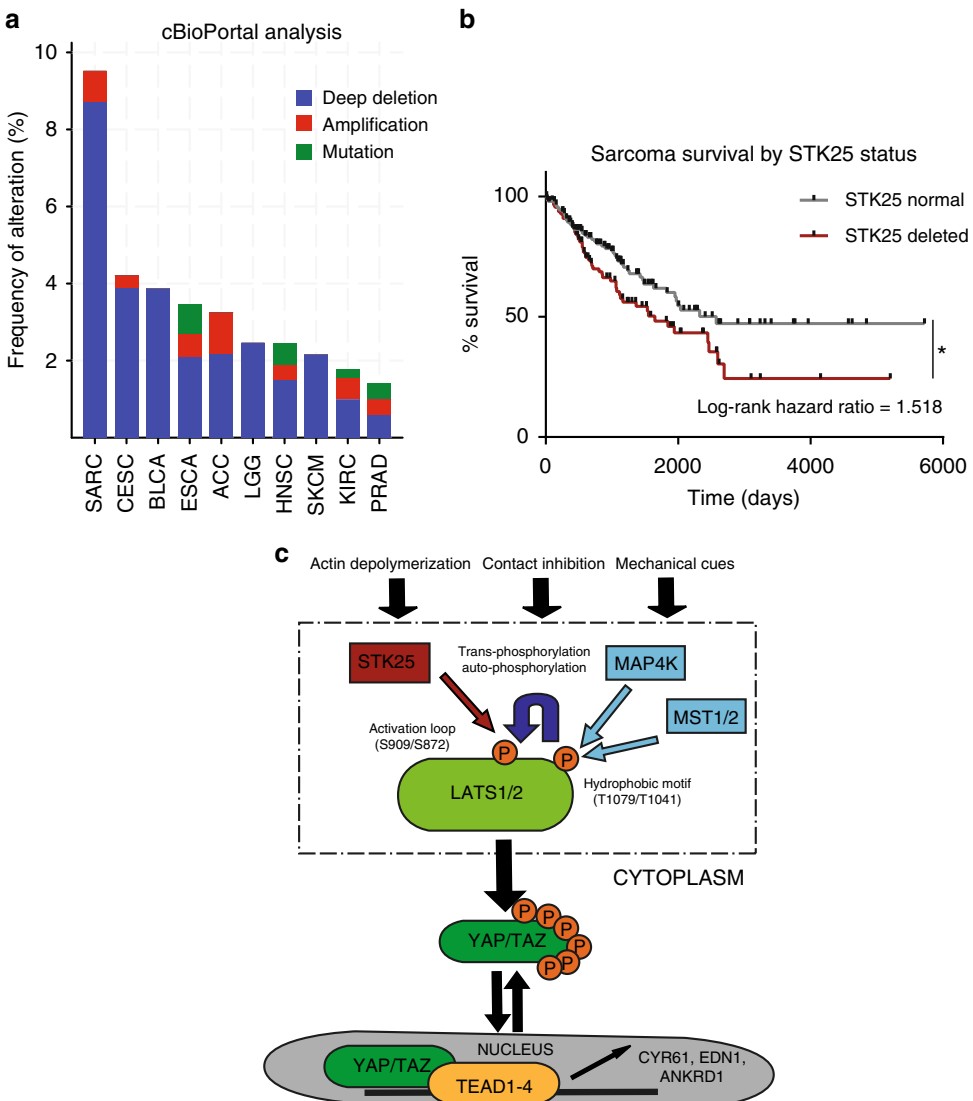

**Fig. 7** *STK25* loss is common in human cancers and adversely affects patient survival. **a** Graphical representation of human cancers with the highest frequencies of *STK25* deletion. Data was accessed using the cBioPortal online program (http://www.cbioportal.org/). **b** Survival data of sarcoma patients from the TCGA dataset were accessed using the Xenabrowser online program (https://xenabrowser.net/) and overall survival rates and times were assessed for patients with and without deletions of *STK25* (*$p = 0.0172$, $n = 215$, log-rank test). **c** Proposed model of STK25 in Hippo tumor suppressor signaling

**Protein immunoblotting and immunoprecipitation.** Cells were rinsed twice with ice-cold 1X PBS (Boston Bioproducts) and lysed immediately with 1X cell lysis buffer (2% w/v SDS, 10% glycerol, 60 mM Tris–HCl) supplemented with 1X HALT protease and phosphatase dual inhibitor cocktail (ThermoFisher). Cell lysates were then sonicated for 15 s at 20 kHz and sample buffer (Boston Bioproducts) was added to a final concentration of 1×, after which protein samples were incubated at 95 °C for 5 min. For immunoprecipitation experiments, cells were rinsed twice with ice-cold PBS as before and lysed with a standard IP lysis buffer (20 mM Tris–HCl, 150 mM NaCl, 1 mM EDTA, 1 mM EGTA, 1% Triton X-100, 1 mM β-glycerophosphate, 1 mM sodium orthovanadate) supplemented with a protease inhibitor tablet (Sigma-Aldrich). Lysates were then transferred to microfuge tubes and centrifuged at 17,000×g for 10 min to collect insoluble pellets. The supernatants containing soluble proteins were used for downstream applications. For protein extraction from mouse liver tissue samples, the tissue samples were briefly rinsed with ice-cold PBS. The tissue samples were then homogenized using a dounce homogenizer (Ambion) and protein was extracted using RIPA tissue lysis buffer (50 mM Tris–HCl, 150 mM NaCl, 1% NP-40 equivalent, 1% sodium deoxycholate, 1 mM β-glycerophosphate, 1 mM sodium orthovanadate, 10 mM sodium fluoride, 0.1% SDS) supplemented with 2X HALT phosphatase inhibitor cocktail (ThermoFisher) and a protease inhibitor tablet (Sigma-Aldrich). Further homogenization of tissue lysates were performed using a 20-gauge needle and brief sonication for 20 s at 20 kHz. The tissue lysates were then centrifuged at 17,000×g for 10 min and supernatants containing the soluble protein were used for downstream

applications. Cell lysates were resolved via SDS-PAGE and transferred to PVDF membranes (Bio-Rad) using a TransBlot Turbo semi-dry transfer system (Bio-Rad) or a wet-tank transfer system (Bio-Rad). Following transfer, membranes were blocked in TBS-0.5% Tween-20 (10 mM Tris–HCl, 150 mM NaCl, 0.5% Tween-20) containing 5% non-fat dried milk (NFDM) for 1 h, and then incubated with primary antibodies diluted in 1% NFDM TBS-0.5% Tween-20 solution. For in vitro kinase assays, membranes were blocked in 5% BSA in TBS-0.1% Tween-20 for 1 h prior to incubation with primary antibodies, which were diluted in 1% BSA TBS-0.1% Tween-20 solution. The antibodies used for quantitative immunoblotting were the following: anti-LATS1 (1:1000, Cell Signaling Technology, 3477), anti-LATS2 (1:500, Cell Signaling Technology, 5888), anti-LATS2 (1:1000, Cell signaling Technology, 13646), anti-LATS2 (1:1000, Bethyl Labratories, A300-479A), anti-YAP (1:1000, Cell Signaling Technology, 14074), anti-P-YAP S127 (1:1000, Cell Signaling Technology, 13008), anti-P-YAP S397 (1:1000, Cell Signaling Technology, 13619), anti-STK25 (1:5000, Abcam, ab157188), anti-MST1 (1:1000, Cell Signaling Technology, 3682), anti-STK3/MST2 (1:5000, Abcam, ab52641), anti-GAPDH (1:5000, Cell Signaling Technology, 2118), anti-P-LATS T1079 (1:500, Cell Signaling Technology 8654), anti-P-LATS S909 (1:500, Cell Signaling Technology, 9157), anti-P-LATS2-S872 (1:500, Signalway Antibodies, 12875), anti-PCNA (1:1000, Cell Signaling Technology, 13110), anti-TAZ (1:1000, BD Biosciences, 560235), anti-FLAG (1:1000, Sigma, P2983), anti-MAP4K1 (1:5000, Abcam, ab33910), anti-β-actin (1:5000, Cell Signaling Technology, 4970), and anti-α-Tubulin (1:10000, Millipore, CP06-100UG). Primary antibodies were

detected using horseradish peroxidase-conjugated species-specific secondary antibodies (1:5000, Cell Signaling Technology) and ECL Prime (GE Amersham), Clarity ECL blotting substrate (Bio-Rad) or Clarity Max ECL blotting substrate (Bio-Rad). Imaging of blots were performed using the ChemiDoc XRS+ imaging system (Bio-Rad), and quantitative densitometry was performed using the Bio-Rad ImageLab software. For immunoprecipitations, proteins of interest were precipitated using Protein G magnetic beads (New England Biolabs) and either anti-FLAG (Sigma-Aldrich, P2983), anti-HA (Abcam, ab18181) or anti-Myc (ThermoFisher, MA1-33272) primary antibodies. To immunoprecipitate endogenous LATS2 from mouse liver lysates, anti-LATS2 antibody (Cell Signaling Technology, 13646) and Protein A magnetic beads (New England Biolabs) were used according to manufacturer's instructions. Following pulldown of protein targets, magnetic beads were washed three times with ice-cold IP lysis buffer and then eluted using 2X sample buffer by incubating at 95 °C for 5 min. Eluted proteins were then assessed via SDS–PAGE. Phos-tag electrophoresis was performed according to manufacturer's instructions, using the manganese reagent option (Wako Chemicals, AAL-107). Uncropped digital scans of the most important immunoblots are presented in Supplementary Figure 8.

**In vitro kinase assay**. Cells transfected with either FLAG-STK25, FLAG-MAP4K1, FLAG-MST1, or HA-LATS2 were lysed in ice-cold modified cell lysis buffer (25 mM Tris–HCl, 150 mM NaCl, 1% NP-40 equivalent, 5% glycerol, 1 mM β-glycerophosphate, 1 mM sodium orthovanadate, 10 mM sodium fluoride) supplemented with a protease inhibitor tablet and 1X HALT phosphatase inhibitor cocktail, and incubated on ice for 30 min with occasional agitation. In order to ensure that the immunoprecipitated LATS2 would be in its un-phosphorylated state at the time of protein extraction, cells transfected with HA-LATS2 were stimulated with 10% FBS for 1 h prior to protein collection. The supernatant containing soluble proteins was collected via centrifugation, and immunoprecipitation was performed as described above. Pellets containing immunoprecipitated proteins of interest were washed three times with ice-cold modified cell lysis buffer, and then three times with ice-cold in vitro kinase assay buffer (20 mM Tris–HCl, 10 mM β-glycerophosphate, 1 mM sodium orthovanadate, 20 mM sodium fluoride, 10 mM magnesium chloride, 1 mM DTT) supplemented with a protease inhibitor tablet. In vitro kinase assays were then performed by mixing the pellets containing immunoprecipitated FLAG-STK25 or FLAG-MAP4K1 together with pellets containing immunoprecipitated HA-LATS2 in a microfuge tube and incubating the mixture at 30 °C for 30 min in the presence of 100 or 500 μM ATP. For reactions involving HA-LATS2 and FLAG-MST1, reactions were performed in a HEPES-based kinase buffer (30 mM HEPES, 50 mM potassium acetate, 10 mM magnesium chloride) for 30 min at 30 °C in the presence of 100 μM ATP. Reactions were terminated by the addition of 4X sample buffer and incubating the mixture at 95 °C for 5 min, and samples were assessed via SDS–PAGE and immunoblotting for LATS phosphorylation.

**Immunofluorescence microscopy**. Cells were washed once with PBS and fixed with 4% paraformaldehyde (Electron Microscopy Sciences) for 15 min. Cells were then extracted with TBS-0.5% Triton X-100 for 5 min, blocked in TBS–BSA (10 mM Tris–HCl, 150 mM NaCl, 5% bovine serum albumin, 0.2% sodium azide) for 30 min at room temperature, and incubated with primary antibodies diluted in TBS–BSA for 1 h in a humidified chamber. The following antibodies and reagents were used: anti-YAP 63.7 (1:250, Santa Cruz Biotechnology, sc-6864), anti-YAP65 (1:200, Abcam, 2060-1), anti-α-Tubulin (1:1000, Millipore, CP06-100UG), and anti-FLAG (1:250, Sigma, P2983). Bound primary antibodies were visualized using species-specific fluorescent secondary antibodies (1:250, Molecular Probes, A11001, 11005, A11008), while DNA was visualized using 2.5 μg/mL Hoechst; F-actin was visualized using rhodamine-conjugated phalloidin (1:500, Molecular Probes, R415). Immunofluorescence images were obtained using a Nikon TE2000-E2 inverted microscope. Images were analyzed using NIS-Elements software. For quantifications of YAP subcellular localization, two boxes of equal size were drawn in individual cells: one in the nucleus, and one in the cytoplasm. The mean fluorescence intensity of YAP was measured in these regions of interest and a nuclear: cytoplasmic ratio was determined. All quantifications of immunofluorescent images were performed in a blinded fashion. To assess rescue of tetraploidy-induced cell cycle arrest via an EdU incorporation assay, hTERT-RPE-1 cells were plated onto glass coverslips and transfected with siRNAs of interest. At 24 h post-transfection, cells were treated with 4 μM DCB for 16 h, and then washed four times with cell culture media and allowed to recover for a further 24 h. Cells were then pulsed with 10 μM EdU (ThermoFisher) for 2 h in fresh media and then immediately fixed in 4% paraformaldehyde. EdU incorporation was visualized using the Click-iT EdU Alexa Fluor 488 kit (ThermoFisher, C10337) as per manufacturer's instructions. DNA was stained using 2.5 μg/mL Hoechst as before. Coverslips were imaged and binucleated cells were scored as being positive if EdU signal was present in both nuclei.

**RNA extraction and qRT-PCR**. Total RNA from cultured cells were extracted using the RNeasy kit (Qiagen), and cDNA was generated from RNA using the Superscript III kit and oligo(dT) primers (Invitrogen). For generation of cDNA from mouse liver samples, total RNA was extracted from mouse liver samples using

the RNeasy kit (Qiagen) and a dounce homogenizer (Ambion) according to manufacturer's instructions. To generate cDNA from mouse liver RNA, oligo(dT) and random hexamers were used in equimolar concentration with the Superscript III kit (Invitrogen). Quantitative real-time PCR was performed using SYBR Green reagents in a StepOnePlus system (Applied Biosystems). Please refer to Supplementary Table 5 for a list of all primers for real-time PCR used in this study.

**Microarray analysis and GSEA**. Total RNA was extracted from exponentially growing RPE-1 cells at 72-h post-transfection with the RNeasy kit (Qiagen) and was hybridized onto Affymetrix HG-U133_Plus_2 arrays according to the manufacturer's instructions. For GSEA, a rank-ordered list of genes from the siSTK25 group relative to controls was used as input for preranked GSEA using the meandiv method for normalization and a weighted enrichment statistic.

**Live cell imaging**. To assess rescue of contact inhibition-mediated cell cycle arrest following siRNA transfection, hTERT-RPE-1-FUCCI cells were grown on glass-bottom 12-well tissue culture dishes (Mattek) and transfected with siRNAs of interest. At 24 h post-transfection, imaging began on a Nikon TE2000-E2 inverted microscope equipped with the Nikon Perfect Focus system. The microscope was enclosed within a temperature and atmosphere-controlled environment at 37 °C and 5% humidified CO$_2$. Fluorescence and phase contrast images were captured every 15 min with a ×10 0.5 NA Plan Fluor objective at multiple points for 72 h. All captured images were quantified using NIS-Elements software in a blinded fashion.

**Mouse models**. *Stk25* mutant mice were generated by deletion of exons 4 and 5 and genotyped via PCR. Heterozygous *Stk25* mice were intercrossed to generate wildtype, heterozygous, and homozygous *Stk25* mice. At >24 weeks of age, both male and female mice were humanely euthanized and livers were collected for histological analysis or snap frozen in liquid nitrogen and stored at −80 °C for analysis of protein and gene expression. All mice were housed under pathogen-free conditions at Worcester Polytechnic Institute (WPI). All experiments were conducted following approval from the WPI Institutional Animal Care and Use Committee (IACUC).

**Immunohistochemistry**. Mouse liver samples were dissected out and immediately fixed in 4% paraformaldehyde for 48 h, after which they were preserved in 70% ethanol for a further 48 h at 4 °C. These liver samples were then embedded in paraffin for sectioning and mounting onto glass coverslips. Mounted liver tissue sections were deparaffinized and rehydrated, and antigen retrieval was performed using a citric acid-based antigen retrieval solution (Vector Labs). Endogenous peroxidase activity was quenched in 3% hydrogen peroxide for 10 min, and blocked in 10% normal goat serum (Sigma-Aldrich) for 1 h at room temperature. Primary antibodies were diluted in 10% normal goat serum and incubated overnight at 4 °C (YAP, 1:400, Cell Signaling Technology, 14074). Anti-rabbit SignalStain Boost IHC detection reagent (Cell Signaling Technology) and SignalStain DAB substrate kit (Cell Signaling Technology) were used according to manufacturer's instructions to develop the slides. Counterstaining was performed with hematoxylin (Cell Signaling Technology) wherein dedifferentiation and bluing was performed with acid ethanol and 0.1% sodium bicarbonate solutions, respectively. The slides were then rinsed, dehydrated, and mounted with ProLong Gold Antifade (ThermoFisher Invitrogen). Images were captured on randomly selected points on each slide using the Echo Revolve Hybrid Microscope system at ×40 (Echo Laboratories) and obtained IHC images were subsequently analyzed using the H-DAB program on Fiji/ImageJ.

**Cell proliferation assay**. Each clonal HEK293A cell line was seeded into six-well plates at a density of $2.5 \times 10^4$ cells per well, and one well was trypsinized and counted every 48 h, for a total of five total time points. A total of three independent experiments was performed, and data is presented as mean ± SEM for each clonal cell line.

**Analysis of TCGA datasets**. Focal deletions of STK25 were analyzed using the Tumorscape online program through the Broad Institute (http://portals.broadinstitute.org/tcga/home) using Gene-Centric GISTIC analysis; the dataset used was the "2015-06-01 stddata_2015_04_02_regular peel-off." To assess levels of focal deletion of other identified upstream Hippo kinases, the same dataset was probed for other MSTs, as well as members of the MAP4K family. To assess STK25 alterations in human cancers, TCGA datasets were assessed using the cBioPortal online program (http://www.cbioportal.org/). To assess clinical outcomes in sarcoma patients with deletions in STK25, the TCGA dataset for Sarcomas was assessed and analyzed using the UCSC Xena Functional Genomics Browser (https://xenabrowser.net/). Patients were scored as having deletions of STK25 based on GISTIC-thresholded reads of STK25 gene copy number; if such copy numbers were either −1 or −2, then the patients were scored as having deletions of STK25. Patients with normal copy numbers of STK25 (0) or amplifications (1 or 2) were concatenated into one group for the purposes of survival analysis.

**Quantification and statistical analysis**. All quantitative data have been presented as mean ± SEM, unless otherwise indicated. The number of samples ($n$) represent the number of biologic replicates, except in the case of immunofluorescent image quantifications, in which they represent the number of cells quantified per group of interest over three or more biologic replicates. Prism 7 was used for all statistical analyses. Data were subjected to the D'Agostino–Pearson normality test to determine whether standard parametric tests should be applied to test for significant differences between groups. Data passing the normality test, or data with insufficient data points to conclusively assess normality were assessed via parametric tests of significance. Data failing the normality test were assessed using non-parametric tests of significance. Experiments in which more than two independent groups were assessed simultaneously, ANOVA with the appropriate post-hoc tests of significant differences were applied. Linear associations, where appropriate, were assessed using Pearson's correlation coefficient. For assessment of statistical differences between cellular proliferation curves, a two-way ANOVA with Tukey's post-hoc analysis was performed. Kaplan–Meier survival analysis of sarcoma patients with *STK25* deletions were performed using the log-rank test. Statistical tests with $p$-values of < 0.05 were considered significant.

**Reporting summary**. Further information on experimental design is available in the Nature Research Reporting Summary linked to this article.

## Data availability

Microarray data generated with this study have been deposited under the accession codes GSE119502 and GSE119503. All relevant data are available within the manuscript and its supplementary information or from the authors upon reasonable request.

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

## Acknowledgements

We would like to thank Tohru Matsuki for generating and sharing the *STK25*$^{-/-}$ MEFs, Kun-Liang Guan and members of his lab for technical advice and cell lines, Duojia Pan for reagents, Bob Varelas and Anurag Singh for critical discussions about the work, and the Boston University Experimental Pathology Core for tissue processing. We would also like to thank members of the Ganem lab for detailed comments on the manuscript. Microarray experiments and gene set enrichment analyses were performed with the help of the Boston University Clinical and Translational Science Institute Bioinformatics Group who are supported by a grant from the NIH/NCATS (1UL1TR001430). S.L. is supported by a Medical Student Research Grant from the Melanoma Research Foundation, a Medical Student Research Grant Award from the American Skin Association, and a Predoctoral Fellowship Award from the American Heart Association (19PRE34370004). M.A.V. is supported by a training grant from the NIH/NIGMS (5T32GM008541-20) and an F30 Award from the NCI (1F30CA228388-01). R.J.Q. is supported by a Canadian Institutes of Health Research Doctoral Foreign Study Award. B.W.H. is supported by NIH/NINDS grant NS073662. A.L.M. is supported by NIH/NCI grant R00CA182731 and the Smith Family Awards Program. N.J.G. is a member of the Shamim and Ashraf Dahod Breast Cancer Research Laboratories and is supported by NIH grants CA154531 and GM117150, the Karin Grunebaum Foundation, the Smith Family Awards Program, the Searle Scholars Program, and the Jackie King Young Investigator Award from the Melanoma Research Alliance.

## Author contributions

S.L., H.C., and N.J.G. conceptualized and designed the study. S.L. performed experiments with R.J.Q. for bioinformatic analysis of datasets, with N.H. and A.L.M. for in vivo mouse experiments and with T.M., H.M.M., M.A.V., and I.P.M.M. for all other experiments. S.L. and N.J.G. analyzed the results. B.W.H. provided critical reagents. S.L. and N.J.G. wrote the manuscript. All authors commented on the manuscript.

## Additional information

**Competing interests:** The authors declare no competing interests.

