## [Peer Review File · Nature Communications]

Reviewers' comments:

Reviewer #1 (Remarks to the Author):

In this manuscript, the authors described STK25 as a LATS activating kinase. Knockdown or knockout of STK25 decreased phosphorylation and cytoplasmic localization of YAP, which is mainly controlled by LATS-induced phosphorylation, whereas STK25 overexpression increased YAP phosphorylation. Similarly, transcription activity of YAP is also negatively regulated by STK25. The effect of STK25 on YAP requires LATS1/2, but not MST or MAP4K. Mechanistically, the authors showed that STK25 could directly phosphorylate the activation loop (AL) of LATS2 kinase.

It is well established that LATS is activated by upstream kinases, such as MST1/2 and MAP4Ks that phosphorylate the hydrophobic motif (HM) in LATS to promote LATS autophosphorylation in the AL and therefore LATS kinase activation. The model proposed in this study has potential significance if it proves to be correct. However, the data presented in the current study is not convincing to support the two key conclusions that STK25 phosphorylates LATS AL and plays a significant role in the regulation of LATS and YAP in vivo.

The first major concern is that the biochemical data supporting STK25 directly phosphorylates LATS is weak. Fig. 4d and S5a, addition of ATP in the STK25-KR sample also increased LATS AL phosphorylation, indicating that the increased LATS AL phosphorylation was not due to STK25 kinase activity, likely due to some contamination. Fig. 4e (this is a key experiment), the data are poor quality and not convincing. Again, addition of ATP without STK25 also increased LATS AL phosphorylation. To support that STK25 phosphorylates LATS AL and activates LATS, two critical experiments are needed. First, what percent of LATS AL can be phosphorylated by STK25, but not the STK25-KR, in vitro? Second, does STK25 activate LATS kinase activity in vitro?

The second concern is the biological function of SKT25 in LATS/YAP regulation. One of the best studied functions of the Hippo pathway is its role in liver size control and tumorigenesis. The authors already have STK25 knockout mice (Fig. 5g). They need to show whether STK25 knockout mice have enlarged livers and develop tumors. These phenotypes are predicted if STK25 indeed plays a role in LATS and YAP regulation. For example, deletions of MST, or NF2, or Sav, cause hepatomegaly and liver tumors, both phenotypes are dependent on YAP. In order to conclude that STK25 acts through the Hippo pathway, the authors should show that the STK25 knockout phenotypes are YAP dependent. This is a common practice in Hippo field research. In addition, do the authors have any explanation why mutation of the drosophila GCKIIISTK25, the closest STK25 homolog, does not produce Hippo phenotypes (Dev Cell 25, 507-19, 2013)?

Fig. 1d. The basal YAP phosphorylation without DCB treatment should be included for the SKT25 KO clones. Fig. 1e. The data is rather strange. Expression of STK25 suppressed DCB-induced YAP phosphorylation, results contradict to the model. Furthermore, it is a bit strange why the reconstitution was done in the siRNA knockdown cells instead of the CRPSR knockout cell lines, which were available (Fig. 1d) and should be much cleaner.

Fig. 3d-f. The effect of STK25 overexpression on YAP localization is rather modest. Many previous studies have shown strong effect of MST or LATS overexpression on YAP localization, again indicating that STK25 may not have a major role in LATS-YAP regulation.

Fig. 5c. Because LATS is the proposed direct target of STK25, LATS phosphorylation should be included. Fig. 5g. Phosphorylation of both YAP and LATS should be included.

Fig. S6b, c. The effect of STK5 knockdown on YAP phosphorylation is rather minor. LATS phosphorylation should be included.

Reviewer #2 (Remarks to the Author):

In this study, Lim et al identified and characterized a new Lats kinase, STK25 that regulates Yap/Taz nuclear localization/activity in response to actin cytoskeletal stress as well as under normal growth suppression conditions in cultured cell lines. Using both loss and gain of function analyses, the authors showed that STK25 phosphorylated Lats independent of other upstream Lats kinases including MST1/2 and MAP4Ks. In addition, they provided evidence that STK25 phosphorylates the activation loop (AL) of Lats independent of Lats autophosphorylation, which is distinct from other upstream kinases that phosphorylate a hydrophobic motif (HM) of Lats to promote its autophosphorylation. Finally, by analyzing TCGA data, the authors found that STK25 was deleted in many types of human malignancies. The identification of STK25 as a novel Lats kinase is interesting and the biochemical characterization of STK25 is quite thorough with quantification provided for most of the data although some of the changes are quite subtle. However, the study was almost exclusively carried out using in vitro cultured cell system with the exception of only one experiment (Fig. 5g), raising a serious concern of the physiological relevance of the findings. In fact, mice with STK25 KO are viable (Amrutkar et al., Diabetes 2015), suggesting that STK25 is not essential for Lats activation in vivo, at least, during development.

Major concerns:

1. While analysis of STK25 in vitro is thorough, the authors only did one experiment, i.e., measuring Yap and Taz levels in KO liver to address the in vivo relevance of STK regulation of Lats. It is not clear whether there is any change in liver size. They should compare Lats1/2 phosphorylation, Yap/Taz phosphorylation, Yap/Taz subcellular localization, Yap/Taz target gene expression, and cell proliferation (Ki67) between control and KO livers or other tissues such as intestines where Hippo pathway plays an essential role in the regulation cell proliferation.
2. It is possible that the lack of overall change of organ size (for example, liver size) could be due to redundancy among the upstream kinases. The physiological role STK25 might be revealed under stress conditions (for example, see Cai et al., G&D 2010). Another possibility to reveal a physiological role of STK25 is to study its fly homolog, as fly has less redundant kinases upstream of Lats, and it is relatively easier to carry out double mutant analysis.
3. In the RNAi screen, knockdown of MST3 and MST4 also exhibited reduced Yap phosphorylation (Fig. S1a-d). Since MST3 and MST4 are closely related to STK25 (also called MST5), do they play partially redundant role with STK25 as is the case for MAP4Ks? For example, does knockdown of MST3/4 enhance the defect caused by STK25 knockdown? Can expression of MST3/4 rescue the defect caused by STK25 knockdown?
4. The rescue experiments shown in Fig. 1e is not convincing, as the pYap signals for the STK25-WT expressing cells were overall weak compared with the vector group. From the blots, it looks to this reviewer that the ratio of pYap/Yap for siSTK25+STK25-WT (lane 6) is much lower rather than higher (shown by the graphic) than that for siSTK25+Vector (lane 3). To examine whether STK25 can rescue Yap phosphorylation caused STK25 RNAi. The authors should directly compare siSTK25 +Vector, siSTK25 + STK25-WT, and siSTK25 + STK25-KD.
5. How did the authors validate individual STK25KO clones generated by CRISPR shown in Fig. 2d-f? Do individual STK25KO clones exhibit elevated Hippo pathway target gene expression? From the image shown in Fig 1. 2d and Fig. S2a, d, g, it looks like the overall levels of Yap instead of nuclear levels of Yap increased In STK25 KO HEK293A cells or STK25 -/- MEF cells. Yap was predominantly localized in the nucleus even in the control cells, especially in WT MEF cells, making it difficult to

access Yap nuclear/cytoplasmic distribution. The authors should treat the cells with DCB to determine whether Yap exits the nucleus in control cells but fails to do so in STK25 KO HEK293A cells and STK25 ^{-/-}MEF.

6. It seems that siSTK25 and LATS dKO had very similar effect on Yap nuclear localization (Fig. 3g). How could that be possible given that STK25 mutant mice were viable? What happened to MST1/2 and MAP4Ks in siSTK25 cells? Why could they activate LATS in siSTK25 cells?

7. How did the authors validate MM8KO cells (this reagent was not even validated in the cited paper)? Wild type 293 cells should be used as a control in the experiment shown in Fig. 3j to determine the relative contribution of STK25 and other upstream kinases to LATS activation. The image shown in Fig. 3k is not convincing. The Flag-positive cell pointed by the arrowhead is on top of two other cells, making it hard to assess the Yap nuclear/cytoplasmic distribution. In fact, in the same panel, there is an isolated Flag-positive cell that exhibited Yap nuclear localization similar to Flag-negative cells, raising a concern whether STK25 could activate LATS independent of MST1/2 and MAP4Ks.

8. To demonstrate that STK25 directly phosphorylates LATS, the authors carried out an in vitro kinase assay using immunopurified STK25 as the kinase source and immunopurified LATS as the substrate. However, one cannot rule out that phosphorylation was carried out by another kinase(s) co-purified with STK25 or LATS. This concern is strengthened by the observation that immunopurified STK25-KD could also phosphorylate the activation loop (AL) of LATS (Fig. 4d, Fig. S5a) and both STK25-WT and STK25-KD could phosphorylate the hydrophobic motif (HM) of LATS (Fig. 4e).

9. The authors argued that phosphorylation of LATS AL by STK25 is sufficient to activate STK25 and can bypass the requirement of HM phosphorylation. If so, one would expect that coexpression of LATS with the HM site mutated together with STK25 should rescue Yap phosphorylation defect in LATS dKO cells in the experiment described in Fig. S5b.

10. It is intriguing to observe that STK25 was deleted in so many types of cancer while deletion of other Hippo pathway components was not observed at all given that STK25 mutant mice are viable. It is not clear how many neighboring genes were deleted together with STK25 in individual cases and whether deletion of the neighboring genes rather than STK25 contributed to the tumor phenotypes. If STK25 is so frequently deleted in cancers, it should be possible to identify cancer cell lines in which STK25 was deleted and then restore STK25 expression to determine whether cancer cell proliferation/survival is affected.

Others:

The rationale of using IMR90 cells to do the RNAi screen is not explained. Has this cell line ever been used to study Hippo signaling? If so, reference should be provided.

STK25 ^{-/-} MEF was not described in the Method.

Page 4: ".....and Yap/Taz phosphorylation remained intact in mice lacking MST1/2 (Zhou et 2009, Meng et al., 2015)"

----Zhou et al actually showed that Yap phosphorylation was affected in MST1/2 KO livers although not in MST1/2 KO MEFs. ???????

Page 5: "(Thomson and Sahai 2014)" should 2015

Reviewer #3 (Remarks to the Author):

This manuscript describes the identification of STK25 as a novel LATS activating kinase. LATS is a critical regulator of Hippo signaling and how it is activated has been the subject of considerable study. It is thought to be activated by phosphorylation on a hydrophobic motif (HM) C-terminal to the kinase domain which (when phosphorylated) triggers the kinase to autophosphorylate in the activation loop (AL) of the kinase domain. In total 11 different kinases (MST1/2, 6 MAP4K family members, and TAO1/2/3) are known to phosphorylate LATS on the HM motif in a somewhat redundant manner. The novel contribution of this paper is that it identifies STK25 as an AL motif kinase, a site previously thought to be an autophosphorylation site. The authors show that STK25 promotes YAP phosphorylation by acting through LATS and that it promotes phosphorylation of LATS on the AL but not HM site. They also show that unlike other LATS kinases, STK25 is focally deleted in various cancers. Overall this is novel interesting work that would be of broad interest to the Hippo field. However prior to publication, there are a few points that should be addressed to bolster their model for STK25 function.

Main point

1) The authors propose that STK25 directly phosphorylates LATS on the AL site and that this is sufficient to activate LATS. An alternative model is that STK25 instead stimulates LATS to autophosphorylate on the AL site. To rule out this possibility, the authors isolated kinase dead LATS (LATS-KD) as well as STK25 from cells with LATS1 and LATS2 deleted and carried out in vitro kinase assays. They concluded from these experiments that STK25 can cause AL phosphorylation on LATS-KD. This is a well conceived experiment, but the blot showing that STK25 phosphorylates LATS-KD is not very convincing - it is not clear if there is a band that ran funny or is it a blotch on the blot. Having a repeat of this blot and quantification would make the result more convincing.

2) In the Discussion the authors state "In contrast, STK25-dependent activation of LATS entirely bypasses the need for phosphorylation at the hydrophobic motif, which provides some explanation for the robustness with which loss of this singular kinase is able to blunt LATS activation." This statement seems a bit strong based on the data shown. If the authors want to make this conclusion then I think that they need to show that STK25 can activate a LATS mutant where the phosphorylated threonine at the HM site has been mutated to alanine.

Other points that would improve the manuscript but are not individually essential.

1) The authors do not mention that TAO kinases also activate LATS by phosphorylating it on the HM site, and although they do a good job showing that STK25 can activate LATS in absence of the other LATS-HM kinases, they do not consider TAO. I am not sure if this has to be addressed experimentally since they show pretty convincingly that unlike TAO and the other HM kinases, STK25 causes LATS-AL and not HM phosphorylation, but it should at least be addressed in the text.

2) On the bottom of page 13 the authors state "...that depletion of STK25 significantly impairs the dynamics of YAP phosphorylation under these conditions, such that not only does the overall magnitude of YAP phosphorylation become blunted, but YAP gets phosphorylated more slowly and to a lesser extent in STK25-depleted cells compared to controls (Fig. 5c)." It is not clear what they mean by this from looking at the figure. Perhaps they could be more explicit regarding which data points should be compared.

3) The authors state "...we found increased levels of YAP and TAZ protein in the STK25^{-/-} mice livers compared to their wild-type controls, indicating that loss of STK25 inactivates Hippo signaling under physiologic conditions both in vitro and in vivo (Fig. 5g)." Since increased YAP/TAZ activity is well known to cause increased liver size and cancer, the authors should at least comment on whether either of these phenotypes are observed in these mice.

Responses to Reviewer #1

We thank the reviewer for thorough review, as well as the constructive and highly thoughtful comments.

“The first major concern is that the biochemical data supporting STK25 directly phosphorylates LATS is weak. Fig. 4d and S5a, addition of ATP in the STK25-KR sample also increased LATS AL phosphorylation, indicating that the increased LATS AL phosphorylation was not due to STK25 kinase activity, likely due to some contamination.”

The reviewer correctly points out that our previous data suggested proteins other than STK25 may be phosphorylating LATS at the activation site motif, since co-incubation of immunoprecipitated kinase-dead STK25 with LATS2 still revealed low levels of LATS phosphorylation at the activation loop site. We also appreciated this result, and believed that the most logical explanation was that kinase-dead STK25 was co-immunoprecipitating endogenous WT-STK25, as STK25 is known to bind to itself (Kean et al., 2011). To address this concern, we generated a LATS1/LATS2/STK25 triple knockout cell line to remove all endogenous wild-type STK25, and then repeated all of the *in vitro* kinase assays. Using this approach, we found that expression of exogenous WT-STK25, but not kinase-dead STK25, was able to significantly promote activation loop phosphorylation of exogenously expressed kinase-dead LATS2 (n=4 biological replicates). This has now been added to revised Figure 4e.

“Fig. 4e (this is a key experiment), the data are poor quality and not convincing.”

We agree that our original *in vitro* kinase assays could be improved to be more convincing. We therefore performed extensive optimization experiments to improve the overall quality of our *in vitro* kinase reactions, including optimization of reaction length, buffer composition, immunoprecipitation conditions, and ATP concentration, some of which we present below. While all optimization conditions gave the same basic result (that STK25 promotes phosphorylation of LATS at the activation site motif and not the hydrophobic motif), we were able to find conditions that maximized the effect and therefore produced the most convincing blots.

This first blot demonstrates that a 1hr incubation promotes the most robust LATS phosphorylation by STK25, while the second blot demonstrates that the kinase assay works best in Tris-HCL buffer with 100uM ATP. We previously had been using 500 uM ATP.

Following optimization of our reaction conditions as presented above, we repeated our experiments in biological quadruplicate, and found that wild-type STK25, but not kinase-dead STK25, was able to significantly induce phosphorylation of kinase-dead LATS2 at the activation loop (Revised Figure 4e). We have revised the methods to include these changes.

“Again, addition of ATP without STK25 also increased LATS AL phosphorylation. To support that STK25 phosphorylates LATS AL and activates LATS, two critical experiments are needed. First, what percent of LATS AL can be phosphorylated by STK25, but not the STK25-KR, in vitro?”

We believe that the improved *in vitro* kinase assay confidently demonstrates the ability of wild-type STK25 to promote LATS-AL phosphorylation, and that our quantifications address the question regarding to what extent kinase-dead STK25 is able to promote LATS-AL phosphorylation relative to STK25-WT (revised Figure 4e). We do observe some basal phosphorylation of LATS-AL in all conditions upon the addition of ATP, although it should be noted that this trend never rises to statistical significance. It is certainly possible that additional kinases are also able to promote LATS-AL phosphorylation, which we now note in the text, but this observation ultimately does not detract from our primary conclusion that STK25 promotes phosphorylation of the LATS activation loop.

“Second, does STK25 activate LATS kinase activity in vitro?”

This is an excellent question. To address this, we opted to use LATS1/2 KO cells (LATS dKO 293A cells) in which we co-expressed hydrophobic motif mutated LATS2^{T1041A} together with STK25-WT, STK25-KD, or MST1. We found that co-expression of STK25-WT with LATS2^{T1041A} increased YAP phosphorylation roughly two-fold over that induced by LATS2^{T1041A} alone, and that neither STK25-KD nor MST1 co-expression with LATS2^{T1041A} could promote further YAP phosphorylation. These data revealed that STK25 robustly activates LATS2^{T1041A} to induce YAP phosphorylation, even in the total absence of a preceding hydrophobic motif phosphorylation (Figure 4h) (n=4 biological replicates).

“The second concern is the biological function of STK25 in LATS/YAP regulation. One of the best studied functions of the Hippo pathway is its role in liver size control and tumorigenesis. The authors already have STK25 knockout mice (Fig. 5g). They need to show whether STK25 knockout mice have enlarged livers and develop tumors. These phenotypes are predicted if STK25 indeed plays a role in LATS and YAP regulation. For example, deletions of MST, or NF2, or Sav, cause hepatomegaly and liver tumors, both phenotypes are dependent on YAP. In order to conclude that STK25 acts through the Hippo pathway, the authors should show that the STK25 knockout phenotypes are YAP dependent.”

We agree with the reviewer that further characterization of STK25^{-/-} mice was needed. To this end, we aged a cohort of STK25^{+/+} and STK25^{-/-} mice to 6-12 months of age and assessed whether the KO mice presented with phenotypes typical to Hippo inactive genetic knockout models. Indeed, we found that STK25^{-/-} mice had moderate, but significant enlargement of their livers (Revised Figure 6a, b), including

one mouse at approximately 11 months of age with massive hepatomegaly (Supplementary Figure 7a). This is in line with other single-gene knockout models of Hippo regulators. Moreover, using a combination of immunoblotting, immunohistochemistry, and qRT-PCR, we confirmed that deletion of *STK25*^{-/-} significantly dampens activation loop phosphorylation of LATS kinases *in vivo*, thus promoting YAP/TAZ activation and enhanced cellular proliferation. These data are in the new Figure 6.

We believe that additional aging past 1 year will be necessary to assess whether loss of *STK25* is sufficient to promote tumorigenesis, and even then, results may prove complex: we observed two ~10-month-old *STK25*^{-/-} mice die during our initial experiments. One appeared to die of lymphoma, while the other had an uncharacterized tumor mass in the chest cavity (possible sarcoma). Both mice also showed enlarged livers. Thus, we believe that long-term cancer studies (including whether effects are YAP/TAZ-dependent) will require significant exploration and are beyond the scope of the current manuscript.

“In addition, do the authors have any explanation why mutation of the drosophila GCKIII/STK25, the closest STK25 homolog, does not produce Hippo phenotypes (Dev Cell 25, 507-19, 2013)?”

We believe that the main reason why *STK25* loss presents with phenotypes more reminiscent of *Hpo* loss is due to evolutionary divergence in mammalian cells. Indeed, there are three kinases that are evolutionarily related to the *Drosophila* GCKIII, which are MST3, MST4, and *STK25*. Interestingly, based on several algorithms, such as DIOPT, *STK25* appears to be most distinct from GCKIII among the three kinases, while MST3 most closely resembles GCKIII. Interestingly, other groups have already reported that modulation of MST3 does not appear to affect Hippo signaling, partly explaining why GCKIII loss does not phenocopy loss of *Hpo* or *Wts* (Zheng et al. 2015, Meng et al. 2015).

“Fig. 1d. The basal YAP phosphorylation without DCB treatment should be included for the SKT25 KO clones.”

Thank you for the suggestion. We now show basal YAP phosphorylation in *STK25* KO clones under conditions of both DMSO and DCB treatment, which is in line with our other experimental observations (Supplementary Figure 1h).

“Fig. 1e. The data is rather strange. Expression of STK25 suppressed DCB-induced YAP phosphorylation, results contradict to the model. Furthermore, it is a bit strange why the reconstitution was done in the siRNA knockdown cells instead of the CRPSR knockout cell lines, which were available (Fig. 1d) and should be much cleaner.”

We thank the reviewer for this comment, and agree the rescue experiment is much cleaner in the *STK25* CRISPR-KO cell lines. We performed rescue experiments in *STK25* KO cells by expressing Cas9-resistant forms of wild-type *STK25* or kinase-dead *STK25*. As expected, we found that reconstitution of wild-type, but not kinase-dead, *STK25* was sufficient to restore global YAP phosphorylation in *STK25* KO cells (Revised Figure 1e). We also performed a phos-tag electrophoresis experiment to demonstrate global YAP phosphorylation levels in our *STK25* KO cells. All experiments were done in biological quadruplicate and quantified.

“Fig. 3d-f. The effect of STK25 overexpression on YAP localization is rather modest. Many previous studies have shown strong effect of MST or LATS overexpression on YAP localization, again indicating that STK25 may not have a major role in LATS-YAP regulation.”

We respectfully disagree with this interpretation, as Figures 3d-f demonstrate that overexpression of *STK25* is able to significantly reduce YAP nuclearity, as well as reduce the proportion of cells with nuclear YAP. We are extremely confident in this quantitative immunofluorescence data, as we always quantitate hundreds of cells, over at least 3 biological replicates, in a completely blinded manner.

Further, this IF data is also consistent with our western blotting data showing that expression of WT-STK25, but not kinase-dead STK25, promotes YAP phosphorylation, even in the MM8KO cell line (Figure 3k, Supplementary Figure 3a,b). We also found that this same overexpression of wild-type STK25 is able to reduce the expression of *CTGF* and *CYR61* by >50% (Supplementary Figure 3d), which, when combined with our other data, strongly supports our finding that STK25 plays an important role in regulating Hippo signaling.

“Fig. 5c. Because LATS is the proposed direct target of STK25, LATS phosphorylation should be included. Fig. 5g. Phosphorylation of both YAP and LATS should be included. Fig. S6b, c. The effect of STK5 knockdown on YAP phosphorylation is rather minor. LATS phosphorylation should be included.”

We have now amended Figure 5g to show that phosphorylation of the activation loop of LATS, but not the hydrophobic motif, is significantly decreased in contact inhibited cells following KO of STK25 (experiment performed in biological triplicate). This experiment is analogous to the experiment performed in Supplementary Figure 6b-d, where we show total phosphorylation of YAP and TAZ. Due to technical challenges involving detection of phosphorylation of the activation loop of LATS (which frequently requires IP of LATS prior to blotting), we were unable to successfully blot for p-LATS in the trypsinized samples held in suspension for various timepoints shown in Figure 5c. However, we now present new data in which we demonstrate that phosphorylation of the activation loop of LATS is decreased in the livers of *STK25*^{-/-} mice, which is much more relevant (revised Figure 6d).

Responses to Reviewer #2

We thank the reviewer for the thoughtful and constructive comments, especially regarding the *in vivo* work.

“1. While analysis of STK25 in vitro is thorough, the authors only did one experiment, i.e., measuring Yap and Taz levels in KO liver to address the in vivo relevance of STK regulation of Lats. It is not clear whether there is any change in liver size. They should compare Lats1/2 phosphorylation, Yap/Taz phosphorylation, Yap/Taz subcellular localization, Yap/Taz target gene expression, and cell proliferation (Ki67) between control and KO livers or other tissues such as intestines where Hippo pathway plays an essential role in the regulation cell proliferation.”

We completely agree with the reviewer's assessment that the *in vivo* relevance of STK25 should be better characterized. To address these concerns, we aged a cohort of *STK25*^{+/+} and *STK25*^{-/-} mice to 6-12 months of age. Indeed, we found a moderate, but significant, increase in liver size in *STK25*^{-/-} mice relative to wild-type controls, which is in line with other single-gene knockout models of Hippo regulators (Revised Figure 6a, b, Supplementary Figure 6a). We also confirmed that deletion of *STK25* significantly dampens activation loop phosphorylation, but not hydrophobic motif phosphorylation, of LATS kinases *in vivo* by immunoblotting protein lysates from livers (Figure 6d, e). In addition to demonstrating that YAP and TAZ levels increase in the livers of *STK25*^{-/-} mice by immunoblotting and immunohistochemistry (Figure 6d, g, Supplementary Figure 6b), we also show that YAP phosphorylation at S397 is also significantly decreased. While we were unable to reliably quantify nuclear localization of YAP/TAZ in livers, we did perform qRT-PCR of well described YAP/TAZ target genes (*INHBA*, *BIRC3*, *CYR61*, and *CTGF*). These data demonstrated that YAP/TAZ target genes were significantly upregulated in *STK25*^{-/-} livers relative to controls (Figure 6c). Finally, in lieu of assessing Ki67 for cell proliferation, we opted to use PCNA levels, which is also a well-established marker of proliferation. We found that *STK25*^{-/-} mice had significantly increased levels of PCNA protein expression in their livers when compared to wild-type control mouse livers, which is in line with our finding that *STK25*^{-/-} mice have increased liver weights (Revised Figures 6d, f). Taken together, we believe that these data demonstrate an important *in vivo* role for STK25 in the regulation of Hippo signaling.

“2. It is possible that the lack of overall change of organ size (for example, liver size) could be due to redundancy among the upstream kinases. The physiological role *STK25* might be revealed under stress conditions (for example, see Cai et al., G&D 2010). Another possibility to reveal a physiological role of *STK25* is to study its fly homolog, as fly has less redundant kinases upstream of *Lats*, and it is relatively easier to carry out double mutant analysis.”

We would like to thank the reviewer for the suggestions, but as described above, we do observe an increased liver size in *STK25*^{-/-} mice.

“3. In the RNAi screen, knockdown of *MST3* and *MST4* also exhibited reduced *Yap* phosphorylation (Fig. S1a-d). Since *MST3* and *MST4* are closely related to *STK25* (also called *MST5*), do they play partially redundant role with *STK25* as is the case for *MAP4Ks*? For example, does knockdown of *MST3/4* enhance the defect caused by *STK25* knockdown? Can expression of *MST3/4* rescue the defect caused by *STK25* knockdown?”

We did see reductions in P-YAP following siRNA-mediated depletion of *MST3* and *MST4* in IMR90 cells as the reviewer correctly points out. However, this effect seemed to be cell-type specific, as we did not see the same result in other cell types, such as HEK293A. Moreover, co-depletion of *MST3/4* was not able decrease YAP phosphorylation in HEK293A cells (presented below), nor did it appear to enhance the decrease in YAP phosphorylation caused by *STK25* knockdown.

Thus, while *MST3/4* may have potentially interesting functions in regulating YAP in certain cellular contexts, we felt it was best to keep the current manuscript focused on *STK25*. We do now cite a recent paper by Poon et al. that demonstrates *MST3/4/STK25* are downstream of TAO kinases in a so-called “Hippo-like” signaling pathway, suggesting that at these kinases might play redundant roles in certain contexts, which we’ve added to our discussion section (Poon et al. 2018).

“4. The rescue experiments shown in Fig. 1e is not convincing, as the pYap signals for the *STK25*-WT expressing cells were overall weak compared with the vector group. From the blots, it looks to this reviewer that the ratio of pYap/Yap for si*STK25*+*STK25*-WT (lane 6) is much lower rather than higher (shown by the graphic) than that for si*STK25*+Vector (lane 3). To examine whether *STK25* can rescue Yap phosphorylation caused *STK25* RNAi. The authors should directly compare si*STK25* +Vector, si*STK25* + *STK25*-WT, and si*STK25* + *STK25*-KD.”

While we are convinced of our data based on our quantitation of P-YAP/YAP in our previous rescue experiment set-up, we agree that the experiment was complicated due to the addition of DCB-treated samples. We have now performed a simpler and cleaner rescue experiment in a CRISPR-generated STK25 KO cell line. We expressed Cas9-resistant forms of wild-type STK25 or kinase-dead STK25 in this line. As expected, we found that reconstitution of wild-type, but not kinase-dead, STK25 was sufficient to restore global YAP phosphorylation in STK25 KO cells (Revised Figure 1e). We also performed a phos-tag electrophoresis experiment to demonstrate global YAP phosphorylation levels in our STK25 KO cells. All experiments were done in biological quadruplicate and quantified.

“5. How did the authors validate individual STK25KO clones generated by CRISPR shown in Fig. 2d-f? Do individual STK25KO clones exhibit elevated Hippo pathway target gene expression?”

Individual STK25 KO clones were validated by immunoblotting for loss of STK25 signal. We do find that individual STK25 KO clones exhibit increased expression of YAP-target genes, which we've added as a supplementary figure (Supplementary Figure 2j).

“From the image shown in Fig 1. 2d and Fig. S2a, d, g, it looks like the overall levels of Yap instead of nuclear levels of Yap increased in STK25 KO HEK293A cells or STK25 -/- MEF cells. Yap was predominantly localized in the nucleus even in the control cells, especially in WT MEF cells, making it difficult to access Yap nuclear/cytoplasmic distribution.”

We are extremely confident that the ratio of nuclear:cytoplasmic YAP increases in STK25-depleted cells, even when basal levels of nuclear YAP in controls are high. We always quantitate hundreds of cells, over at least 3 biological replicates, in a completely blinded manner for these experiments. To do this, we measure the fluorescence intensity of YAP in a defined area in the cytoplasm and then compare that value to the fluorescence intensity of YAP in the same sized area in the nucleus of the same cell. In fact, we believe that our quantitation of nuclear:cytoplasmic YAP far exceeds the normal standard in the field (most groups simply assess nuclear:cytoplasmic enrichment by eye). For all immunofluorescence localization experiments, we always show the corresponding quantitation. As for YAP levels, we do frequently see stabilization of YAP in STK25-depleted cells (e.g. YAP levels are increased in STK25^{-/-} livers (Figure 6d)).

“The authors should treat the cells with DCB to determine whether Yap exits the nucleus in control cells but fails to do so in STK25 KO HEK293A cells and STK25 -/- MEF.”

Unfortunately, we were unable to find treatment conditions in which we could reliably assess DCB-treated MEFs for YAP localization, because DCB treatment caused the MEFs to round and detach from coverslips. Moreover, we felt that immunofluorescent localization of YAP in STK25 KO cells following DCB treatment (analogous to the experiment with siSTK25 in HEK293A) would be redundant and not add anything substantive to our paper. That being said, we did assess whether or not STK25 KO cells exhibited a global decrease in YAP phosphorylation compared to control cells following DCB treatment via phos-tag electrophoresis, which has been added as a supplementary figure (Supplementary Figure 1h).

“6. It seems that siSTK25 and LATS dKO had very similar effect on Yap nuclear localization (Fig. 3g). How could that be possible given that STK25 mutant mice were viable?”

It is true that acute loss of STK25 strongly promotes nuclear localization of YAP in cell culture, similar to loss of LATS1/2. However, YAP nuclear localization represents only one, indirect assessment of YAP activity. In addition, data acquired in cell culture systems are fundamentally different than *in vivo* models. While we now show that STK25^{-/-} mice show enlarged livers, they clearly do not phenocopy loss of LATS1/2 *in vivo*. This is because loss of STK25 activity reduces, but does not entirely eliminate, LATS activity (Figure 6).

“What happened to MST1/2 and MAP4Ks in siSTK25 cells? Why could they activate LATS in siSTK25 cells?”

We assessed levels of MST1/2 in cells depleted of STK25, and found that they were not perturbed, which has been added as a supplementary figure (Supplementary Figure 1i). We also performed qRT-PCR to check the levels of MAP4Ks in cells depleted of STK25 (presented below) and noted that they were actually increased in some cases, which might be due to cells attempting to compensate for loss of a Hippo activating kinase. This may explain our finding that knockout of STK25 leads to decreased phosphorylation of LATS at the activation loop, but increased phosphorylation of LATS at the hydrophobic motif (Figure 5g). Clearly, STK25 is not required for MST1/2 and MAP4K proteins to phosphorylate LATS at the hydrophobic motif.

“7. How did the authors validate MM8KO cells (this reagent was not even validated in the cited paper)?”

The MM8KO 293A cell line has been published in other peer-reviewed articles since the initial publication (Plouffe et al. 2016, Feng et al. 2016, Meng et al. 2018), such that we are confident of its characterization and validity for studying Hippo signaling. Furthermore, in our opinion, Meng et al. thoroughly characterized MM8KO 293A cells in the initial 2015 publication for the following reasons. First, they assessed MM8KO cells for loss of protein via immunoblotting for MST1, MST2, MAP4K4, MAP4K6, and MAP4K7. Second, they performed surveyor mutation detection assays for MAP4K1, MAP4K2, and MAP4K3, confirming the presence of indel mutations at the sgRNA-corresponding genomic loci. Finally, Meng et al. also phenotypically confirmed that MM8KO 293A had decreased YAP phosphorylation relative to their controls, as well as relative to the MM5KO (MST1/2/MAP4K4/6/7 KO) cells. As such, we are confident of the validity of the MM8KO 293A cell line. We also found that this cell line exhibited significantly decreased YAP phosphorylation compared to control cells, as would be expected. This is now shown in Revised Supplementary Figure 3e.

“Wild type 293 cells should be used as a control in the experiment shown in Fig. 3j to determine the relative contribution of STK25 and other upstream kinases to LATS activation.”

We now include a revised Supplementary Figure 3e, which shows P-YAP levels in control WT HEK293A cells, LATS1/2 dKO cells, and MM8KO 293A cells, all with and without STK25 knockdown. These data reveal that loss of STK25 decreases YAP phosphorylation in WT and MM8KO cells, but not in LATS dKO 293A cells, as we anticipated. Moreover, we find that MM8KO 293A cells have significantly reduced levels of YAP phosphorylation relative to WT 293A controls as mentioned above, which is

further reduced upon knockdown of STK25, demonstrating that loss of STK25 is able to further decrease YAP phosphorylation in the context of MST/MAP4K depletion.

“The image shown in Fig. 3k is not convincing. The Flag-positive cell pointed by the arrowhead is on top of two other cells, making it hard to assess the Yap nuclear/cytoplasmic distribution. In fact, in the same panel, there is an isolated Flag-positive cell that exhibited Yap nuclear localization similar to Flag-negative cells, raising a concern whether STK25 could activate LATS independent of MST1/2 and MAP4Ks.”

We agree with the reviewer’s assessment that this experimental design could be improved. To better assess whether or not STK25 truly acts independently of MSTs/MAP4Ks, we generated MM8KO 293A cells that stably overexpressed either wild-type STK25 or kinase-dead STK25. Using quantitative western blotting (and phos-tag gels), we found that overexpression of wild-type STK25, but not kinase-dead STK25, increased YAP phosphorylation in these cell lines (n=4 biological replicates), demonstrating that STK25 acts independently of MSTs/MAP4Ks. These data have now been added as Revised Figure 3k.

“8. To demonstrate that STK25 directly phosphorylates LATS, the authors carried out in vitro kinase assay using immunopurified STK25 as the kinase source and immunopurified LATS as the substrate. However, one cannot rule out that phosphorylation was carried out by another kinase(s) co-purified with STK25 or LATS. This concern is strengthened by the observation that immunopurified STK25-KD could also phosphorylate the activation loop (AL) of LATS (Fig. 4d, Fig. S5a) and both STK25-WT and STK25-KD could phosphorylate the hydrophobic motif (HM) of LATS (Fig. 4e).”

These are excellent points. As detailed above (in addressing Reviewer 1), we optimized our *in vitro* kinase assay reaction conditions and then performed our experiments in a LATS1/LATS2/STK25 triple KO background to ablate co-immunoprecipitation of endogenous wild-type STK25, which was likely confounding our data. Under these new conditions, we found that only wild-type STK25 was able to significantly promote phosphorylation of kinase-dead LATS2 at the activation loop, which we believe convincingly supports our major statement that STK25 is a novel activator of LATS (Revised Figure 4e). As the reviewer notes, we do observe some basal phosphorylation of LATS-AL in all conditions upon the addition of ATP, although it should be noted that this trend never rises to statistical significance. It is certainly possible that additional kinases are also able to promote LATS-AL phosphorylation, which we now include in the text, but this observation ultimately does not detract from our primary conclusion that STK25 promotes phosphorylation of the LATS activation loop.

“9. The authors argued that phosphorylation of LATS AL by STK25 is sufficient to activate STK25 and can bypass the requirement of HM phosphorylation. If so, one would expect that coexpression of LATS with the HM site mutated together with STK25 should rescue Yap phosphorylation defect in LATS dKO cells in the experiment described in Fig. S5b.”

This was an extremely valuable suggestion. To improve this portion of the manuscript, we generated a non-phosphorylatable threonine to alanine hydrophobic motif mutant of LATS2 (LATS2^{T1041A}), co-expressed it with WT-STK25 in a LATS1/2 KO background, and assessed YAP phosphorylation. Our data revealed that STK25 robustly activates LATS2^{T1041A} to induce YAP phosphorylation, even in the total absence of a preceding hydrophobic motif phosphorylation (Figure 4h) (n=4 biological replicates).

“10. It is intriguing to observe that STK25 was deleted in so many types of cancer while deletion of other Hippo pathway components was not observed at all given that STK25 mutant mice are viable. It is not clear how many neighboring genes were deleted together with STK25 in individual cases and whether deletion of the neighboring genes rather than STK25 contributed to the tumor phenotypes.”

According to Tumorscape, there are 18 other genes in the *STK25* focally deleted peak. While none are known tumor suppressors, we certainly cannot rule out that some of these additional genes also play roles in tumor development. We have included a supplementary table including these gene names (Revised Supplementary Figure 7c).

“If STK25 is so frequently deleted in cancers, it should be possible to identify cancer cell lines in which STK25 was deleted and then restore STK25 expression to determine whether cancer cell proliferation/survival is affected.”

We attempted to reconstitute wild-type *STK25* in cancer cells harboring deletions or kinase-dead mutations of *STK25*, including MFE-296 (an endometrial cancer cell line with a heterozygous deletion of *STK25*) and HCT116 (colorectal cancer cell line with G160W putative kinase-dead mutation in *STK25*). However, while we were able to generate cancer cell lines stably overexpressing both pWZL Vector and pWZL *STK25*-KD, we were unable to generate any cell lines expressing *STK25*-WT, suggesting that its presence is toxic to these cells. However, since these experiments consisted of overexpression of *STK25*, rather than simple restoration of physiological levels of *STK25*, and that we could not assess Hippo activity in these cells due to death, we do not feel comfortable including this data in the current manuscript.

“The rationale of using IMR90 cells to do the RNAi screen is not explained. Has this cell line ever been used to study Hippo signaling? If so, reference should be provided.”

We have added references regarding usage of IMR90 cells in studying Hippo signaling to the main text (Stein et al. 2015, Xie et al. 2013).

“STK25 -/- MEF was not described in the Method.”

Thank you for pointing this out. We have now added this protocol to our methods section.

“Page 4: “.....and Yap/Taz phosphorylation remained intact in mice lacking MST1/2 (Zhou et 2009, Meng et al., 2015)”. Zhou et al actually showed that Yap phosphorylation was affected in MST1/2 KO livers although not in MST1/2 KO MEFs.”

Thank you for this correction. We have made the appropriate changes to our text.

“Page 5: “(Thomson and Sahai 2014)” should 2015.”

We have corrected this error. Thank you.

Responses to Reviewer #3

We thank the reviewer for the encouraging and enthusiastic comments regarding the significance of our work, and for several insightful suggestions.

“1) The authors propose that STK25 directly phosphorylates LATS on the AL site and that this is sufficient to activate LATS. An alternative model is that STK25 instead stimulates LATS to autophosphorylate on the AL site. To rule out this possibility, the authors isolated kinase dead LATS (LATS-KD) as well as STK25 from cells with LATS1 and LATS2 deleted and carried out in vitro kinase assays. They concluded from these experiments that STK25 can cause AL phosphorylation on LATS-KD. This is a well conceived experiment, but the blot showing that STK25 phosphorylates LATS-KD is not very convincing - it is not clear if there is a band that ran funny or is it a blotch on the blot. Having a repeat of this blot and quantification would make the result more convincing.”

We thank the reviewer for this comment, and for suggesting quantification as a method. As detailed above for the other two reviewers, we have optimized our *in vitro* kinase reaction conditions and repeated our experiments using proteins immunopurified from a LATS1/2-STK25 triple KO background to reduce contamination from co-immunoprecipitating endogenous wild-type STK25. Upon analysis of a quadruplicate of such *in vitro* kinase reactions, we found that only wild-type STK25 was able to significantly promote phosphorylation of the activation loop on kinase-dead LATS2. This data is presented in figure 4 (Revised Figure 4e).

“2) In the Discussion the authors state “In contrast, STK25-dependent activation of LATS entirely bypasses the need for phosphorylation at the hydrophobic motif, which provides some explanation for the robustness with which loss of this singular kinase is able to blunt LATS activation.” This statement seems a bit strong based on the data shown. If the authors want to make this conclusion then I think that they need to show that STK25 can activate a LATS mutant where the phosphorylated threonine at the HM site has been mutated to alanine.”

We agree that this was a very strong assertion, and that it required further experimental substantiation. To improve this portion of the manuscript, we generated a non-phosphorylatable threonine to alanine hydrophobic motif mutant of LATS2 (LATS2^{T1041A}), co-expressed it with WT-STK25 in a LATS1/2 KO background, and assessed YAP phosphorylation. Our data reveal that STK25 robustly activates LATS2^{T1041A} to induce YAP phosphorylation, even in the total absence of a preceding hydrophobic motif phosphorylation (Figure 4h) (n=4 biological replicates). We would like to thank the reviewer for this extremely helpful suggestion.

“1) The authors do not mention that TAO kinases also activate LATS by phosphorylating it on the HM site, and although they do a good job showing that STK25 can activate LATS in absence of the other LATS-HM kinases, they do not consider TAO. I am not sure if this has to be addressed experimentally since they show pretty convincingly that unlike TAO and the other HM kinases, STK25 causes LATS-AL and not HM phosphorylation, but it should at least be addressed in the text.”

We thank the reviewer for this comment. We now mention the potential interactions between TAO and STK25, especially in light of recent findings from *Drosophila* indicating that Tao and GckIII interact as part of a signaling cascade (Poon et al. 2018).

“2) On the bottom of page 13 the authors state “...that depletion of STK25 significantly impairs the dynamics of YAP phosphorylation under these conditions, such that not only does the overall magnitude of YAP phosphorylation become blunted, but YAP gets phosphorylated more slowly and to a lesser extent in STK25-depleted cells compared to controls (Fig. 5c).” It is not clear what they mean by this from looking at the figure. Perhaps they could be more explicit regarding which data points should be compared.”

We have now simplified this text to better describe observations found in Figure 5c. We now write, “We also grew adherent cells in suspension for defined periods of time, as cell detachment is another known activator of LATS activity (Zhao et al. 2012). We found that depletion of STK25 reproducibly impaired phosphorylation of YAP under conditions of cell detachment, consistent with our other findings regarding the role of STK25 in regulating YAP phosphorylation (Fig. 5c).”

“3) The authors state “...we found increased levels of YAP and TAZ protein in the STK25-/- mice livers compared to their wild-type controls, indicating that loss of STK25 inactivates Hippo signaling under physiologic conditions both in vitro and in vivo (Fig. 5g).” Since increased YAP/TAZ activity is well known to cause increased liver size and cancer, the authors should at least comment on whether either of these phenotypes are observed in these mice.”

We have performed extensive experimental work to further characterize the *STK25*^{-/-} mice, as detailed above. We do observe increased liver weights in these KO mice, consistent with other published single gene-knockout mouse models of Hippo signaling. This data is presented in revised Figure 6.

References for Response to Reviewers

1. Feng X, et al. 2016. Thromboxane A2 activates YAP/TAZ protein to induce vascular smooth muscle cell proliferation and migration. *J. Biol. Chem.* **291**: 18947-18958.
2. Kean M.J. et al. 2011. Structure-function analysis of core STRIPAK proteins: a signaling complex implicated in Golgi polarization. *J Biol. Chem.* **286**: 25065-25075.
3. Meng, Z. et al. 2015. MAP4K family kinases act in parallel to MST1/2 to activate LATS1/2 in the Hippo pathway. *Nat Commun.* **6**:8357.
4. Meng Z, et al. 2018. RAP2 mediates mechanoresponses of the Hippo pathway. *Nature.* **560**: 655-660.
5. Plouffe, S.W. et al. 2016. Characterization of Hippo Pathway Components by Gene Inactivation. *Mol Cell* **64**:993-1008.
6. Poon, C.L.C. et al. 2018. A Hippo-like signaling pathway controls tracheal morphogenesis in *Drosophila melanogaster*. *Dev. Cell.* **47**: 564-575.
7. Zhang, N. et al. 2010. The Merlin/NF2 tumor suppressor functions through the YAP oncoprotein to regulate tissue homeostasis in mammals. *Dev. Cell.* **19**: 27-38.
8. Zheng, Y. et al. 2015. Identification of Happyhour/MAP4K as Alternative Hpo/Mst-like Kinases in the Hippo Kinase Cascade. *Dev Cell.* **34**:642-55.

REVIEWERS' COMMENTS:

Reviewer #1 (Remarks to the Author):

The authors have made good efforts to address my concerns. However, the major problem is that STK25 KO does not produce a convincing Hippo phenotype. The new data in Fig. 6b is rather peculiar. The difference in liver size appears simply due to one big liver outlier in the STK25^{-/-} mice. It should be noted that LATS1/LAT2 knockout produced very strong phenotypes that even liver specific knockout mice die before weaning due to lack of functional hepatocytes. Therefore, this study has not convincingly established that STK25 plays a significant physiological role in LATS activation.

A few minor points.

In new Fig.4e, why STK25-KD caused a mobility shift on the band detected by p-LATS-AL? In addition, there are less LATS2 and STK25 in lane 6 than lane 4. Thus, it is difficult to assess whether STK25-WT was more potent than STK25-KD to increase LATS phosphorylation. An important missing control is a reaction without the LATS substrate.

The new data in Fig. 4h is not very conclusive as LATS2 protein was also increased in the STK25 transfected cells. Further complicating the authors' interpretation is the fact that STK25 associates with STRIPAK, which is known to interact with and inactivate MST1/2. So, STK25 overexpression could have an indirect effect on the Hippo pathway.

Reviewer #2 (Remarks to the Author):

In the revision, the authors adequately addressed most of my concerns by conducting additional experiments (e.g. by showing the increased liver size in STK25 mutant mice) and improving the quality of several key experimental data set. Therefore, I recommend publication.

Reviewer #3 (Remarks to the Author):

The revised manuscript contains a number of new experiments that significantly bolster the conclusions and enhance the impact of the manuscript. Overall the authors have done a good job addressing various concerns from the initial review. However prior to publication, there are a few points that should be addressed.

1) The authors show that STK25 (unlike MST1/2) can activate a LATS2 mutant where the threonine at the HM site has been mutated to alanine (LATS2-T/A). However LATS2-T/A that has been phosphorylated by STK25 is only as active as wild-type LATS2 that has not been activated by MST1/2 or other HM kinases. Thus it is not clear how STK25 activated LATS2 compares to wild-type LATS2 activated by MST1/2 or one of the other HM kinases. It would have been nice if the authors had tested this in Figures S4G and 4H. I expect that based on some of their data they could estimate how the activity of LATS2-T/A (activated by STK25) compares to that of wild-type LATS2 (activated by MST1/2). I do not think that it is critical that they repeat the experiments in S4G and 4H, but it would be informative if they could at least give an estimate of the relative activity. Given the apparent reduced ability of STK25 (compared to MST1/2) to activate LATS2, perhaps the authors could discuss whether they think STK25 acts alone or in combination with an HM kinase *in vivo*.

2) I noted in the first review the TAO kinases also activate LATS by phosphorylating it on the HM site, and although the authors showed that STK25 can activate LATS in absence of the other LATS-HM kinases, they do not consider TAO. I did not feel that this needed to be addressed experimentally

since they show pretty convincingly that unlike TAO and the other HM kinases, STK25 causes LATS activation loop phosphorylation. They should note that TAO is an HM kinase that they did not consider. In the revised manuscript the authors now mention TAO as potentially acting upstream of STK25, but not that it is an HM kinase. This should be corrected.

3) I think that Figure 4A lacks appropriate controls and should be removed because it is redundant with Figure 4B, which has controls.

4) The authors state that YAP-S127 phosphorylation in the liver is under redundant control by both MST1/2 and LATS1/2 and cite Zhou et al 2009. This is not what this reference shows. It shows that YAP-S127 phosphorylation in the liver requires MST1/2 but not LATS1/2. This should be corrected. (note that the data in this manuscript is nevertheless consistent with that in Zhou et al.)

5) The sources and sequences of the siRNA used should be listed.

Responses to Reviewer #1

We thank the reviewer for the insightful comments regarding our revised manuscript. We provide a point-by-point response below.

The authors have made good efforts to address my concerns. However, the major problem is that STK25 KO does not produce a convincing Hippo phenotype. The new data in Fig. 6b is rather peculiar. The difference in liver size appears simply due to one big liver outlier in the STK25^{-/-} mice.

The reviewer points out that one outlier in our STK25^{-/-} mice does increase the overall difference between the wild-type mice and the KO mice. However, when we remove the outlier data-point and re-analyze the data, we find that the difference between our wild-type and KO livers still remains significant at $p=0.017$, demonstrating that the outlier is not the driver of this observation.

It should be noted that LATS1/LATS2 knockout produced very strong phenotypes that even liver specific knockout mice die before weaning due to lack of functional hepatocytes. Therefore, this study has not convincingly established that STK25 plays a significant physiological role in LATS activation.

We do not believe that loss of *STK25* should phenocopy genetic ablation of both *LATS1* and *LATS2*. We observe a decrease in Hippo signaling in the liver, rather than a complete loss of Hippo signaling. This is entirely in line with other mouse models of decreased Hippo signaling, such as deletion of *MST1/2* (Zhou et al., 2009) or *Nf2* (Zhang et al. 2010), which do not phenocopy loss of *LATS1/2*. To highlight these differences, we have addressed differences between our mouse model and *LATS1/2* KO mouse model in the text, citing relevant work (Lee et al. 2016).

In new Fig.4e, why STK25-KD caused a mobility shift on the band detected by p-LATS-AL? In addition, there are less LATS2 and STK25 in lane 6 than lane 4. Thus, it is difficult to assess whether STK25-WT was more potent than STK25-KD to increase LATS phosphorylation. An important missing control is a reaction without the LATS substrate.

We do observe what appears to be a mobility shift on our p-LATS-AL blot in the STK25-KD co-incubation group; however, we believe that this observation does not detract from our overall interpretation that STK25-WT strongly promotes phosphorylation at the LATS-AL in all of our reactions. We do note in our revised manuscript that although only STK25-WT was able to significantly promote the phosphorylation of LATS-AL, we do observe residual trends towards increased p-LATS-AL in the other lanes as well upon addition of ATP, suggesting that there may be other, yet to be identified kinases that promote phosphorylation of LATS-AL in a similar fashion to STK25.

The new data in Fig. 4h is not very conclusive as LATS2 protein was also increased in the STK25 transfected cells. Further complicating the authors' interpretation is the fact that STK25 associates with STRIPAK, which is known to interact with and inactivate MST1/2. So, STK25 overexpression could have an indirect effect on the Hippo pathway.

We respectfully disagree with this interpretation. Although the reviewer is correct in pointing out that there appears to be more LATS2^{T1041A} protein in the STK25-WT co-transfected cells (when compared to Vector co-transfected cells), we do not believe that these increases in LATS2 protein levels are responsible for the increased YAP phosphorylation. First, we see similarly elevated levels of LATS2^{T1041A} signal in the STK25-KD co-transfected cells, but we do not see an increase in levels of phosphorylated YAP, suggesting that it is the presence of active STK25-WT that is promoting this additional phosphorylation of YAP when compared to expression of LATS2^{T1041A} alone. Second, we do not believe that the association of STK25 with the STRIPAK complex confounds interpretation of our data, as the mechanism by which the STRIPAK complex acts is via inhibition of MST1/2, and thus

depends on the LATS hydrophobic motif (Bae et al. 2017, Zheng et al. 2017). Our experiment in Fig. 4h utilized LATS2^{T1041A} expressed in a LATS dKO background, so that we would be able to assess if STK25 acted on LATS-YAP signaling in a manner independent of the presence of hydrophobic motif phosphorylation. Consistent with this experimental design, when we overexpress LATS2^{T1041A} together with MST1, which activates LATS via hydrophobic motif phosphorylation, we do not see increased YAP phosphorylation beyond that induced by expression of LATS2^{T1041A} alone. As such, we do not believe that STK25 association with STRIPAK, although interesting and worthy of additional experimental follow-up, is influencing our experimental results as presented in Fig. 4h, which is consistent with the rest of our data which supports an alternative mechanism of activation for LATS kinases by STK25, in which STK25 directly promotes phosphorylation of the LATS-AL.

Responses to Reviewer #2

In the revision, the authors adequately addressed most of my concerns by conducting additional experiments (e.g. by showing the increased liver size in STK25 mutant mice) and improving the quality of several key experimental data set. Therefore, I recommend publication.

We thank the reviewer for his/her advice and comments, which we believe greatly enhanced the overall quality of our work.

Responses to Reviewer #3

We thank the reviewer for the thoughtful comments. We are glad to hear that our follow-up experimental work was found to bolster our initial findings and enhanced the quality of our work. We provide a point-by-point response to the final points below.

1) The authors show that STK25 (unlike MST1/2) can activate a LATS2 mutant where the threonine at the HM site has been mutated to alanine (LATS2-T/A). However LATS2-T/A that has been phosphorylated by STK25 is only as active as wild-type LATS2 that has not been activated by MST1/2 or other HM kinases. Thus it is not clear how STK25 activated LATS2 compares to wild-type LATS2 activated by MST1/2 or one of the other HM kinases. It would have been nice if the authors had tested this in Figures S4G and 4H. I expect that based on some of their data they could estimate how the activity of LATS2-T/A (activated by STK25) compares to that of wild-type LATS2 (activated by MST1/2). I do not think that it is critical that they repeat the experiments in S4G and 4H, but it would be informative if they could at least give an estimate of the relative activity. Given the apparent reduced ability of STK25 (compared to MST1/2) to activate LATS2, perhaps the authors could discuss whether they think STK25 acts alone or in combination with an HM kinase in vivo.

We thank the reviewer for pointing this observation out. Our manuscript demonstrated that STK25 is able to activate a mutant version of LATS2 in which the hydrophobic motif phosphorylation site is mutated to an alanine (LATS2^{T1041A}), thereby demonstrating that STK25 is able to induce YAP phosphorylation in the total absence of a preceding hydrophobic motif phosphorylation event. Interestingly, we noted in this experiment that although this hydrophobic motif mutant LATS2 was able to phosphorylate YAP, its ability to induce such YAP phosphorylation was roughly half that of wild-type LATS2, suggesting that STK25 most likely acts in concert with other LATS-HM-phosphorylating kinases to regulate the activity of LATS. When we compare this observation together with our finding that deletion STK25 in livers results in a compensatory increase in LATS-HM phosphorylation, we believe that this suggests STK25 as playing a cooperative role with other LATS-HM kinases *in vivo*. These points have been added to the discussion.

2) I noted in the first review the TAO kinases also activate LATS by phosphorylating it on the HM site, and although the authors showed that STK25 can activate LATS in absence of the other LATS-HM kinases, they do not consider TAO. I did not feel that this needed to be addressed experimentally since they show pretty convincingly that unlike TAO and the other HM kinases, STK25 causes LATS

activation loop phosphorylation. They should note that TAO is an HM kinase that they did not consider. In the revised manuscript the authors now mention TAO as potentially acting upstream of STK25, but not that it is an HM kinase. This should be corrected.

We thank the reviewer for pointing this out. We have now added a comment indicating that TAO is a LATS-HM kinase in the main text.

3) The authors state that YAP-S127 phosphorylation in the liver is under redundant control by both MST1/2 and LATS1/2 and cite Zhou et al 2009. This is not what this reference shows. It shows that YAP-S127 phosphorylation in the liver requires MST1/2 but not LATS1/2. This should be corrected. (note that the data in this manuscript is nevertheless consistent with that in Zhou et al.)

Thank you for pointing this out. We have corrected our manuscript accordingly.

5) The sources and sequences of the siRNA used should be listed.

Thank you for this comment. We initially submitted this table as a supplementary file in the first draft of our manuscript. This table has now been added to the supplementary materials section.

References for Response to Reviewers

1. Zhang, N. et al. The Merlin/NF2 tumor suppressor functions through the YAP oncoprotein to regulate tissue homeostasis in mammals. *Dev. Cell.* **19**, 27-38 (2010).
2. Zhou, D. et al. Mst1 and Mst2 maintain hepatocyte quiescence and suppress hepatocellular carcinoma development through inactivation of the Yap1 oncogene. *Cancer Cell.* **19**, 425-438 (2009).
3. Bae SJ et al. SAV1 promotes Hippo kinase activation through antagonizing the PP2A phosphatase STRIPAK. *eLife.* **6**, e30278 (2017).
4. Zheng Y et al. Homeostatic control of Hpo/MST kinase activity through autophosphorylation-dependent recruitment of the STRIPAK PP2A phosphatase complex. *Cell Rep.* **21**, 3612-3623 (2017).